# The metabolic stress-activated checkpoint LKB1-MARK3 axis acts as a tumor suppressor in high-grade serous ovarian carcinoma

Hidenori Machino[1,2,3], Syuzo Kaneko [1✉], Masaaki Komatsu[1,2], Noriko Ikawa[1], Ken Asada [1,2], Ryuichiro Nakato[4], Kanto Shozu [1], Ai Dozen[1], Kenbun Sone[3], Hiroshi Yoshida[5], Tomoyasu Kato[6], Katsutoshi Oda[3,7], Yutaka Osuga[3], Tomoyuki Fujii[3], Gottfried von Keudell[8], Vassiliki Saloura[9] & Ryuji Hamamoto [1,2✉]

High-grade serous ovarian carcinoma (HGSOC) is the most aggressive gynecological malignancy, resulting in approximately 70% of ovarian cancer deaths. However, it is still unclear how genetic dysregulations and biological processes generate the malignant subtype of HGSOC. Here we show that expression levels of microtubule affinity-regulating kinase 3 (*MARK3*) are downregulated in HGSOC, and that its downregulation significantly correlates with poor prognosis in HGSOC patients. MARK3 overexpression suppresses cell proliferation and angiogenesis of ovarian cancer cells. The LKB1-MARK3 axis is activated by metabolic stress, which leads to the phosphorylation of CDC25B and CDC25C, followed by induction of G2/M phase arrest. RNA-seq and ATAC-seq analyses indicate that MARK3 attenuates cell cycle progression and angiogenesis partly through downregulation of AP-1 and Hippo signaling target genes. The synthetic lethal therapy using metabolic stress inducers may be a promising therapeutic choice to treat the LKB1-MARK3 axis-dysregulated HGSOCs.

[1] Division of Medical AI Research and Development, National Cancer Center Research Institute, 5-1-1 Tsukiji, Chuo-ku, Tokyo 104-0045, Japan. [2] Cancer Translational Research Team, RIKEN Center for Advanced Intelligence Project, 1-4-1 Nihonbashi, Chuo-ku, Tokyo 103-0027, Japan. [3] Department of Obstetrics and Gynecology, Graduate School of Medicine, The University of Tokyo, Bunkyo, Tokyo 113-8655, Japan. [4] Institute for Quantitative Biosciences, The University of Tokyo, Bunkyo, Tokyo 113-0032, Japan. [5] Division of Diagnostic Pathology, National Cancer Center Hospital, 5-1-1 Tsukiji, Chuo-ku, Tokyo 104-0045, Japan. [6] Department of Gynecology, National Cancer Center Hospital, 5-1-1 Tsukiji, Chuo-ku, Tokyo 104-0045, Japan. [7] Division of Integrative Genomics, Graduate School of Medicine, The University of Tokyo, Bunkyo, Tokyo 113-8655, Japan. [8] Memorial Sloan-Kettering Cancer Center, New York, NY 10065, USA. [9] Center for Cancer Research, National Cancer Institute, Bethesda, MD 20892, USA. ✉email: sykaneko@ncc.go.jp; rhamamot@ncc.go.jp

High-grade serous ovarian carcinoma (HGSOC) is the most aggressive gynecological malignancy, often detected at a late clinical-stage due to its rapid dissemination and metastasis, causing ~70% of deaths from ovarian cancer[1]. Given that *TP53* mutations are ubiquitously observed in both HGSOCs and their precursor tumors, serous tubal intraepithelial carcinomas (STICs), it is likely that TP53 inactivation occurs at the initial step of tumorigenesis[2]. In addition, since TP53 inactivation induces chromosomal instability, almost all HGSOCs have been genetically characterized by a low frequency of point mutations, and high frequency of copy number alterations (CNAs). Therefore, gene expression-based research is prioritized over gene mutation-based research to elucidate the biology of HGSOCs[3–5]. However, the current therapeutic strategy for HGSOCs is only focused on small numbers of well-characterized gene alterations such as *BRCA1* and *BRCA2* (*BRCA1/2*) mutations, suggesting that there remain opportunities to explore other therapeutic targets in HGSOCs by gene expression-based research[6].

HGSOCs are phenotypically classified into two distinct subtypes: homologous recombination (HR)-deficient type and HR-proficient type. About a half of HGSOCs belong to the HR-deficient type, in which HR genes, such as *BRCA1/2*, *RAD51C*, *ATM*, *CHEK2*, or Fanconi anemia genes are genetically or epigenetically inactivated. HR deficiency renders cancer cells vulnerable to DNA double-strand breaks; hence, the platinum-based drugs and poly ADP-ribose polymerase (PARP) inhibitors, causing stalled replication forks and subsequent mitotic catastrophe, are highly effective for this subtype[6,7]. On the other hand, the remaining half of HGSOCs fall into the HR-proficient type, maintaining the function of the HR pathway. Therefore, they often exhibit primary resistance to both platinum-based drugs and PARP inhibitors, leaving a substantial number of untreatable cases[4,7].

Among HR-proficient types, *CCNE1* amplification is the most frequently observed genomic alteration, which accelerates the G1/S phase transition, and results in poor clinical outcomes[8]. This indicates that cell cycle dysregulation is critical to developing HR-proficient HGSOCs. However, it should be noted that forced entry into mitosis may result in hazardous consequences because cell cycle transition is rigidly controlled by cell cycle checkpoints. For example, the ATR–CHEK1 axis phosphorylates CDC25C at serine 216 upon DNA damage; also, the p38–MAPKAPK2 axis phosphorylates CDC25B at serine 323 upon DNA damage or metabolic stress, causing G2/M phase arrest or cell death. Thus, cancer cells must escape from these cell cycle checkpoints if they are to proceed into mitosis under stress-saturated conditions[9,10].

Indeed, accumulating evidence indicates that HGSOCs abolish the G1/S phase checkpoint by inactivating TP53 and RB1 signaling and amplifying Cyclin family genes[3,11]. In contrast, there is little data that explains how HGSOCs successfully escape from the G2/M phase checkpoint. The ATR–CHEK1 axis, a major DNA damage-activated G2/M phase checkpoint, is often upregulated to maintain genomic integrity in HGSOCs. We hypothesize that there is a hidden molecular mechanism enabling HGSOCs to evade the G2/M phase checkpoint, independent of DNA damage-activated stress response.

Microtubule affinity-regulating kinase 3 (MARK3) is a serine/threonine kinase that belongs to the AMPK-related kinase family. MARK3 is reportedly activated by tumor suppressor liver kinase B1 (LKB1) and antagonizes oncogenic pathways, including cell cycle pathway via CDC25C phosphorylation[12–14]. In the present study, it was reported that the LKB1–MARK3 axis is a metabolic stress-activated G2/M phase checkpoint with a mode of action different from that of the ATR–CHEK1 axis, the DNA damage-activated G2/M phase checkpoint, and that the molecules involved in the LKB1–MARK3 axis are highly dysregulated in HGSOCs. These findings may explain how the dysfunction of the LKB1–MARK3 axis results in proliferative HGSOCs in the presence of the DNA damage-activated G2/M phase checkpoint. Metabolic stress inducers may be promising therapeutic choices for LKB1–MARK3 axis-dysregulated cancers.

## Results

**Integrative analysis to identify potential therapeutic target genes in HGSOCs.** Gene expression-based screening was initiated to explore therapeutic target genes in HGSOCs (Fig. 1a). To mitigate batch effects, cross-references to multiple public datasets were conducted via different experimental procedures from independent studies. First, two microarray datasets were analyzed to obtain differentially expressed genes (DEGs) between normal human ovarian surface epithelial cells (HOSEs) and HGSOCs. Overlapped top DEGs ($n = 100$) from two datasets yielded a total of six candidate genes: *BNC1*, *MAF*, *MARK3*, *NKX3-1*, *PDE8B*, and *REEP1* (Fig. 1b–d and Supplementary Fig. 1a). To further validate our analyses, RNA-seq data from TCGA and GTEx projects were surveyed using GEPIA[15], indicating that *MARK3* and *MAF* were consistently downregulated in HGSOCs (Supplementary Fig. 1b, c). Besides, *MARK3* mRNA expression levels were well-correlated with MARK3 protein expression levels (Supplementary Fig. 1d). Survival analysis revealed that the mRNA expression levels of *MARK3* and *BNC1* demonstrated a negative correlation and a positive correlation with the clinical outcome, respectively (Fig. 1e and Supplementary Fig. 1e, f).

Among the DEGs, *BNC1* is reportedly downregulated in several cancer types, such as breast[16], pancreatic[17], hepatocellular[18], and kidney renal cell carcinomas[19] through DNA promoter hypermethylation. Although *MAF* is known for its oncogenic translocation in multiple myeloma[20], when the gene expression levels between cancer tissues and their normal counterparts were compared using GEPIA, *MAF* downregulation, rather than its upregulation was frequently observed. Since MAF is a component of the activating protein-1 (AP-1) complex, that regulates various cellular signaling pathways, including cell differentiation, proliferation, and apoptosis, the dysregulation of *MAF* may cause both oncogenic and tumor-suppressive effects depending on the cancer type-specific environment[21].

MARK3 is a serine/threonine kinase that is directly activated by tumor suppressor LKB1[12] and TAOK1[22], and inactivated by oncogenic PIM1[14]. Substrates of MARK3 cover a wide range of cancer-relevant signal cascades, such as the cell cycle[13], RAS signaling[23], cAMP-PKA signaling[24], JNK signaling[25], and Notch signaling pathways[26]. Although these pieces of evidence suggest tumor-suppressive functions of *MARK3*, there is no consensus on whether *MARK3* is an oncogene or a tumor suppressor gene; additionally, there is no reliable study describing cancer type-specific dysregulation of *MARK3*. Given that our analysis identified that downregulation of *MARK3* is significantly associated with poor clinical outcomes (Fig. 1e) and platinum-resistant status in patients with HGSOC (Fig. 1f), we hypothesized that *MARK3* acts as a tumor suppressor gene and offers a promising therapeutic opportunity.

To validate our in silico analyses in the experimental setting, *MARK3* expression levels across HOSEs, human fallopian tube secretory epithelial cells (HFTSECs), and HGSOCs were compared. RT-qPCR revealed that *MARK3* mRNA expression was indeed downregulated in both ovarian cancer cell lines and clinical tissues of HGSOC (Fig. 1g, h). We also validated that MARK3 expression was decreased at the protein level in several ovarian cancer cell lines (Fig. 1i). Intriguingly, immunohistochemical staining showed that MARK3 was strongly detected in the cell membrane and cytoplasm of HFTSECs, and STICs; however, its presence was diminished in primary HGSOCs,

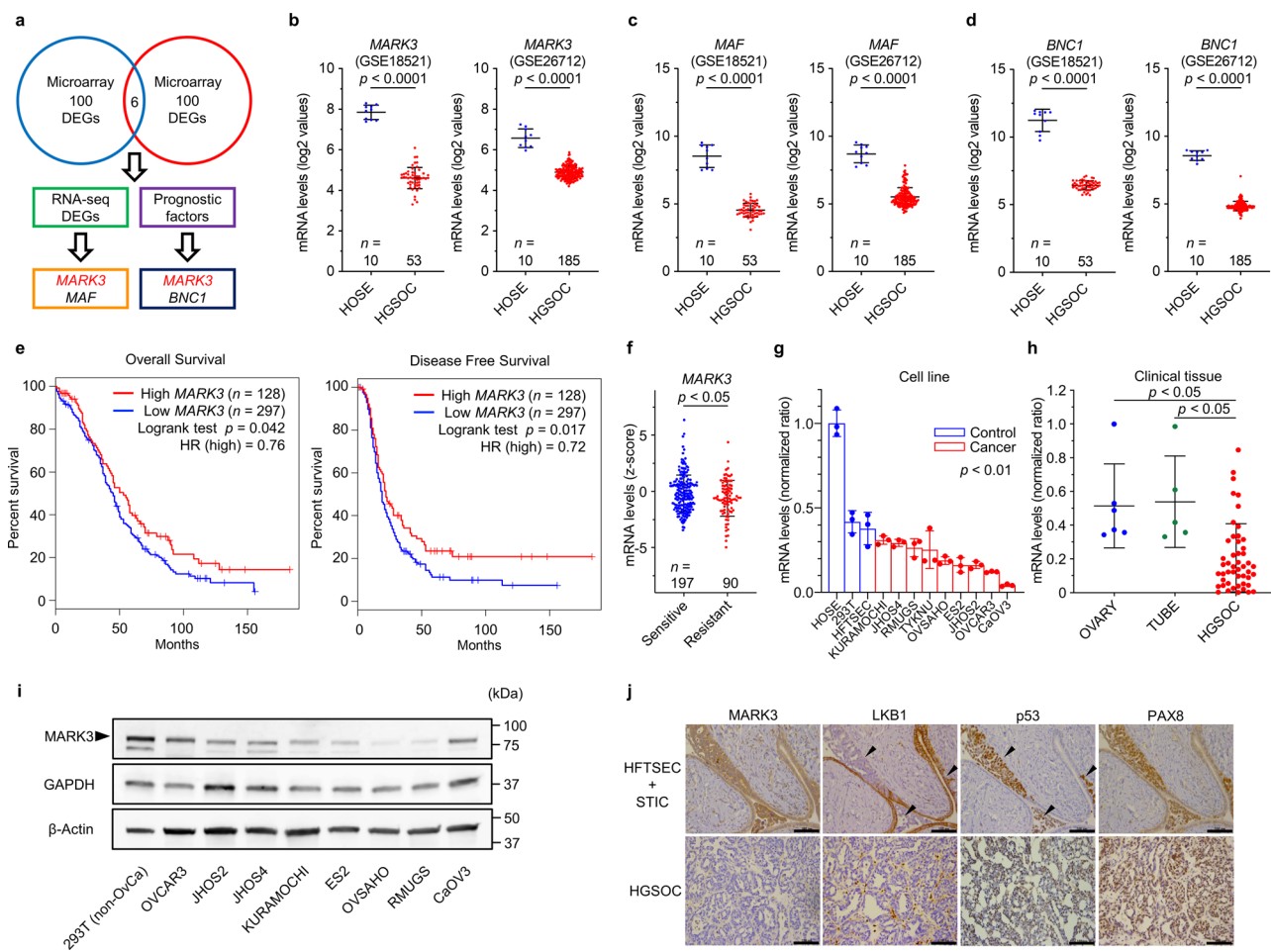

**Fig. 1 Downregulation of *MARK3* is associated with poor clinical outcome in patients with HGSOC. a** Schematic of in silico integrative analysis to identify HGSOC-associated dysregulated genes. *MARK3* is downregulated in HGSOCs across two microarray datasets and one RNA-seq data set, and low *MARK3* mRNA expression is associated with poor clinical outcomes in the TCGA HGSOC cohort. **b–d** mRNA expression levels of *MARK3* (**b**), *MAF* (**c**), and *BNC1* (**d**) in two independent microarray datasets. GSE18521 and GSE26712 offer microarray data of human ovarian surface epithelial cell (HOSE) samples and HGSOC samples [GSE18521(HOSE: $n = 10$, HGSOC: $n = 53$) and GSE26712 (HOSE: $n = 10$, HGSOC: $n = 185$)]. Statistical analysis was performed using an unpaired Student's $t$-test. **e** Kaplan–Meier survival curves classified by high ($n = 128$) or low ($n = 297$) *MARK3* mRNA expression in the TCGA HGSOC cohort. Low *MARK3* group exhibits poor overall survival (left) and poor disease-free survival (right). **f** Platinum-resistant HGSOCs ($n = 90$) have lower *MARK3* mRNA expression compared to platinum-sensitive HGSOCs ($n = 197$). Error bars represent mean ± standard deviation (SD). Statistical analysis was performed using an unpaired Student's $t$-test. **g, h** RT-qPCR shows that *MARK3* expression is downregulated in ovarian cancer cell lines [comparison between control ($n = 3$) and ovarian cancer cell line ($n = 9$)] (**g**) and the HGSOC tissues [ovary ($n = 6$), fallopian tube ($n = 5$) and HGSOC ($n = 49$)] (**h**). Ovarian cancer cell lines with HGSOC-like genetic profile, such as *TP53* mutation and hyper CNA are included in this experiment. Error bars represent mean ± SD from three biological replicates. Statistical analysis was performed using an unpaired Student's $t$-test. **i** Immunoblotting shows that MARK3 expression is downregulated in ovarian cancer cell lines compared to that in 293T cells (non-OvCa; non-Ovarian Carcinoma). **j** Immunohistochemistry displays that MARK3 is ubiquitously expressed in human fallopian tube secretory epithelial cells (HFTSECs) and serous tubal intraepithelial carcinomas (STICs), whereas its expression is diminished in primary HGSOCs. The expression of LKB1 is clearly decreased in STICs (arrow). p53 is used as a positive marker for STICs (arrow) and HGSOCs. Scale bars, 100 μm.

suggesting that the expression of MARK3 was repressed at a later stage of cancer development. On the other hand, the expression of LKB1, an upstream activator of MARK3, was decreased as early as the STICs stage, suggesting the LKB1–MARK3 axis might be gradually undermined during tumorigenesis (Fig. 1j). Collectively, our integrative analyses of public databases, as well as our experiments, demonstrated that *MARK3* might be a promising candidate gene involved in the progression of HGSOCs.

**CNA of *MARK3* and its upstream regulator genes suppresses MARK3 activity**. To elucidate the mechanisms of *MARK3* suppression, the correlation between mRNA expression levels and CNA or DNA promoter methylation profiles was surveyed using the TCGA HGSOC data set, including *MAF* and *BNC1* as

references. Copy number deletion was observed in 38.4% of cases at the *MARK3* loci, in 81.6% of cases at the *MAF* loci, as well as in 48.1% of cases at the *BNC1* loci, which was the dominant CNA in all genes (Fig. 2a). Intriguingly, CNA at the *MARK3* loci was positively correlated with the mRNA expression levels, but that at *MAF* and *BNC1* loci did not significantly influence gene expression levels (Fig. 2b). In contrast, DNA promoter methylation levels of *BNC1* were frequently high, but that of *MARK3* and *MAF* were low across almost all HGSOCs (Supplementary Fig. 2a). These results suggested that mRNA levels of *MARK3* were affected by CNA, whereas those of *BNC1* were affected by DNA promoter methylation.

In addition to the downregulation of gene expression, MARK3 activity could be directly suppressed by upstream regulators such

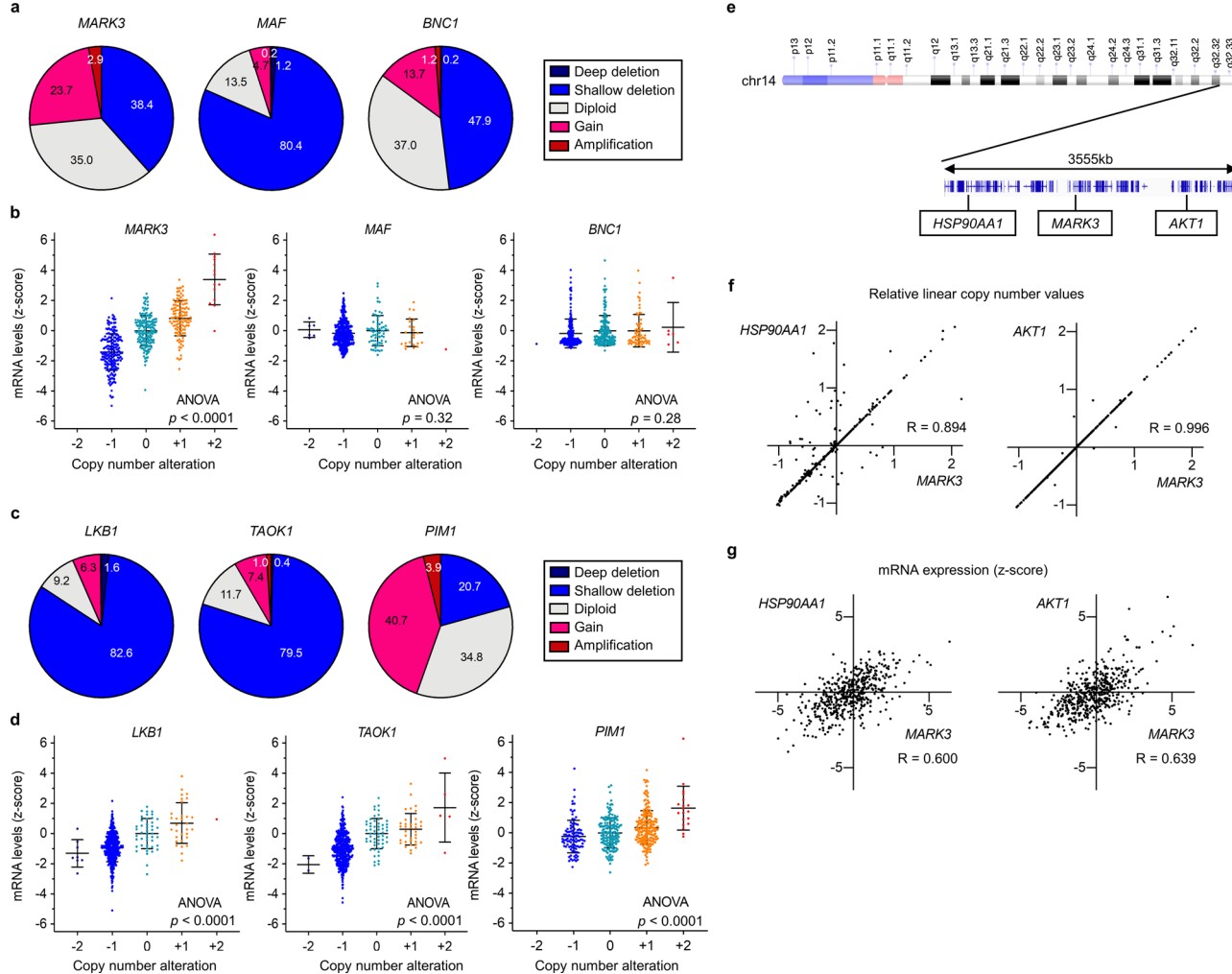

**Fig. 2 Activity and expression of MARK3 are suppressed by genomic and epigenetic alterations. a** Pie charts show putative copy number alteration (CNA) profiles of *MARK3*, *MAF*, and *BNC1*. **b** CNA and mRNA expression levels of *MARK3*, *MAF*, and *BNC1* are plotted using the TCGA HGSOC data set (*n* = 489). Each sample was segregated according to their CNA status: amplification (CNA = +2); gain (CNA = +1); duplicate (CNA = 0); deletion (CNA = −1); deep deletion (CNA = −2). Error bars represent mean ± standard deviation (SD). Statistical analysis was performed using one-way ANOVA. **c** Pie charts show putative CNA profiles of *LKB1*, *TAOK1*, and *PIM1*. **d** CNA and mRNA expression levels of *LKB1*, *TAOK1*, and *PIM1* are plotted using the TCGA HGSOC data set (*n* = 489). Error bars represent mean ± SD. Statistical analysis was performed using one-way ANOVA. **e** Schematic of the 3555-kb long genomic region around *MARK3* loci in chromosome 14q32.33. **f** Relative linear copy number values of *MARK3* are plotted against those of *HSP90AA1* and *AKT1* (*n* = 489). The data were analyzed using Pearson's correlation coefficients. **g** mRNA expression levels of *MARK3* are plotted against those of *HSP90AA1* and *AKT1* (*n* = 489). The data were analyzed using Pearson's correlation coefficients.

as LKB1, TAOK1, and PIM1. It was noted that *LKB1* and *TAOK1*, upstream genes for *MARK3* activation, highly suffered from copy number deletion. In contrast, *PIM1*, an upstream gene modulating MARK3 inactivation, frequently encountered a gain in the copy number (Fig. 2c). CNA profiles and mRNA expression levels were well-correlated in these three genes (Fig. 2d). These observations, in part, explain the mechanism that suppresses MARK3 activity in HGSOCs.

Although our results suggested that MARK3 activity was impaired by both downregulation and inactivation, paradoxically, copy number gain cases at the *MARK3* loci were occasionally observed in HGSOCs (Fig. 2a; 23.7%). This may imply the existence of a selective pressure to increase the copy number at the *MARK3* loci, presumably through the oncogenic demands of neighboring genes. Indeed, putative oncogenes such as *HSP90AA1* (chr14: 102,080,742–102,139,749) and *AKT1* (chr14: 104,769,349–104,795,748) were located around the *MARK3* (chr14: 103,385,394–103,503,831) locus (Fig. 2e). CNAs and mRNA expression levels, between *MARK3* and these oncogenes,

strikingly correlated, indicating that the copy numbers of these genes were simultaneously altered in the majority of HGSOCs (Fig. 2f, g). These results might explain why both copy number deletion and copy number gain occurred at this genomic region in HGSOCs.

To further certify the existence of the trade-off pressure to decrease copy number in this genomic region, the copy number profiles of the *AKT* family (*AKT1*, *AKT2*, and *AKT3*) and *MARK* family (*MARK1*, *MARK2*, *MARK3*, and *MARK4*) were interrogated across a variety of cancer types. Surprisingly, this analysis revealed that the occurrence of *AKT1* amplification was significantly lower than that of *AKT2* and *AKT3*; conversely, the occurrence of the *AKT1* bi-allelic deletion was significantly higher than that of *AKT2* and *AKT3*. However, gene mutation frequency was not different among the three genes (Supplementary Fig. 2b). A similar trend was observed in the *MARK* family; higher bi-allelic deletion was observed at the *MARK3* loci compared to the deletion rate in other *MARK* family members (Supplementary Fig. 2c). These findings suggest that CNAs

around *MARK3* and *AKT1* loci were balanced by both oncogenic and tumor-suppressive demands of the neighboring genes, apparently explaining the reason why copy number gain occurs at the *MARK3* locus.

It was also examined whether epigenetic mechanisms are involved in the downregulation of *MARK3*. The chemical screening was performed using inhibitors targeting epigenetically repressive enzymes. The results showed that the pan-HDAC inhibitors, namely trichostatin A and belinostat, upregulated *MARK3* mRNA expression in a dose-dependent manner, indicating that HDAC activity appears to be involved in repressing *MARK3* mRNA expression (Supplementary Fig. 3a). Last, gene mutation profiles of *MARK3* across 22 cancer types were summarized from the TCGA project. *MARK3* mutations were frequently observed in uterine corpus endometrial carcinoma and skin cutaneous melanoma. Since HGSOC is a cancer type with a low frequency of point mutations, the occurrence of *MARK3* mutations in HGSOCs was restrictive (Supplementary Fig. 3b). It can be hypothesized that MARK3 activity was undermined in HGSOCs via copy number deletion, dysregulation of upstream kinases, and epigenetic repression.

**MARK3 exhibits anti-tumor effects in ovarian cancer cell lines**. To examine tumor-suppressive roles of MARK3 in HGSOCs, a doxycycline (DOX)-inducible system was generated, in which MARK3 was conditionally expressed in OVCAR3, CaOV3, and 293T cells (Supplementary Fig. 4a). As evaluated using the colony formation assay, MARK3 overexpression significantly inhibited cell proliferation of OVCAR3 and CaOV3 (Fig. 3a, b). Cell viability assays showed that the DOX treatment in MARK3-inducible OVCAR3 resulted in a significant decrease in cell number; however, this was not evident in parental OVCAR3 cells (Fig. 3c). Transient overexpression of MARK3 in various ovarian cancer cell lines, JHOS4, CaOV3, RMUGS, OVSAHO and OVCAR3, resulted in decreased cell proliferation (Fig. 3d)[27].

To gain further insights into the molecular mechanism by which MARK3 activity repressed oncogenic signaling pathways, RNA-seq analyses were conducted using MARK3-inducible OVCAR3, yielding downregulated ($n = 1885$, FDR < 0.05) and upregulated genes ($n = 1687$, FDR < 0.05) (Fig. 3e). Pathway analyses using ingenuity pathway analysis (IPA) predicted that MARK3 overexpression resulted in the reduced activity of oncogenes such as *MYC*, *ESR1*, *KRAS*, and *VEGF*, as well as the elevated activity of tumor suppressor genes such as *PTEN*, *CDKN1A*, and *CDKN2A*, supporting the tumor-suppressive roles of MARK3 (Supplementary Data 1). Enrichment analyses by Kyoto Encyclopedia of Genes and Genomes (KEGG) and Gene Ontology (GO) revealed that genes involved in the cell cycle, cell–cell adhesion, and cell proliferation were downregulated (Fig. 3f). These results indicated that *MARK3* played tumor-suppressive roles in modulating several oncogenes and tumor suppressor genes in ovarian cancer cells.

**MARK3 directly phosphorylates CDC25B, which inhibits nuclear translocation of CDC25B**. To elucidate the detailed molecular function of MARK3, potential protein substrates were explored based on kinase recognition sequences. Given that MARK3 phosphorylates CDC25C[13], HDAC7[28], KSR1[23], and PKP2[29] on amino acids within the conserved 14-3-3 binding motif, it was assumed that there was a consensus motif for MARK3 substrates, which coincided with the known 14-3-3 binding motif[30]. On this basis, phosphorylation motif analysis was performed, and it was found that CDC25B serine 323 fit within the MARK3 substrate motif (Fig. 4a). Importantly, CDC25B serine 323

is indeed a 14-3-3 binding site, phosphorylated by checkpoint kinase MAPKAPK2 upon DNA damage or metabolic stress, which promotes the cytoplasmic translocation of CDC25B[10]. Therefore, it was hypothesized that MARK3 acts as a cell cycle checkpoint kinase, which phosphorylates CDC25B at serine 323, as well as the known substrate CDC25C at serine 216.

To certify this hypothesis, a kinase assay was performed using recombinant MARK3 and recombinant CDC25B purified from insect cells. As expected, CDC25B serine 323 was phosphorylated in the presence of MARK3, and this phosphorylation required ATP and magnesium ions, indicating that MARK3 kinase activity could directly phosphorylate CDC25B (Fig. 4b). Overexpression of MARK3 in 293T and OVCAR3 cells resulted in the elevated phosphorylation of CDC25B serine 323 (Fig. 4c). Immunocytochemistry showed the dominant cytoplasmic localization of MARK3 in 293T cells, suggesting that MARK3 mainly functioned in the cytoplasmic environment (Fig. 4d). Indeed, MARK3 overexpression augmented the cytoplasmic localization, and reduced the nuclear localization, of CDC25B. Importantly, alanine substitution of serine 323 in CDC25B (S323A) prevented its cytoplasmic localization in the presence of MARK3, indicating that the subcellular localization of CDC25B was mediated through MARK3-dependent serine 323 phosphorylation (Fig. 4e and Supplementary Fig. 4b). As mentioned previously, MARK3 overexpression also increased the ratio of CDC25C cytoplasmic localization (Fig. 4f), confirming that MARK3 could inhibit nuclear translocation of CDC25B/C.

Since CDC25B/C is an activators of CDK1, which promotes G2/M phase transition, it can be assumed that MARK3 overexpression induces G2/M phase arrest by inhibiting the nuclear translocation of CDC25B/C. Although MARK3 overexpression alone caused no noticeable change in cell cycle progression, there was an increase in the number of cells in the G2/M population following treatment with Ro-3306, a CDK1 inhibitor (Fig. 4g). A cell viability assay for a dose-escalating experiment with a CDK1 inhibitor revealed that DOX-inducible MARK3 overexpression augmented the cytoreductive effect of the CDK1 inhibitor in OVCAR3 cells (Supplementary Fig. 4c). In addition, rescue experiments using mutant CDC25B (S323A) negated the cytoreductive effect of MARK3 (Supplementary Fig. 4d). These results suggested that MARK3 synergistically antagonized CDK1 in response to CDK1 inhibitor treatment during G2/M phase arrest mediated by CDC25B/C phosphorylation (Fig. 4h).

**The LKB1−MARK3 axis is a cytoplasmic cell-cycle checkpoint activated by metabolic stress**. Although stress responses to activate cell cycle checkpoint kinases MAPKAPK2 and CHEK1 have been well documented, activation mechanisms of MARK3 upon stress exposure remain to be described[31]. To elucidate the stress-induced activation mechanism of MARK3, MARK3-inducible 293T cells were treated with various stress inducers: protein synthesis inhibitor (anisomycin), endoplasmic reticulum (ER) stress inducer (thapsigargin), ATP starvation inducer (metformin), oxidative stress inducer (SIN-1), inflammatory cytokine (TNF-α), nutrient starvation (FBS reduction), and hypoxia. Because LKB1 and TAOK1 activate MARK3 via phosphorylation of threonine 211 within the activation T-loop, phosphorylated threonine 211 levels were used to measure MARK3 activation[12,22]. Immunoblotting results showed that MARK3 was activated by anisomycin and thapsigargin (Fig. 5a and Supplementary Fig. 5a). Notably, treatment with DNA damage stress inducers, cisplatin, and doxorubicin, did not activate MARK3 (Fig. 5b).

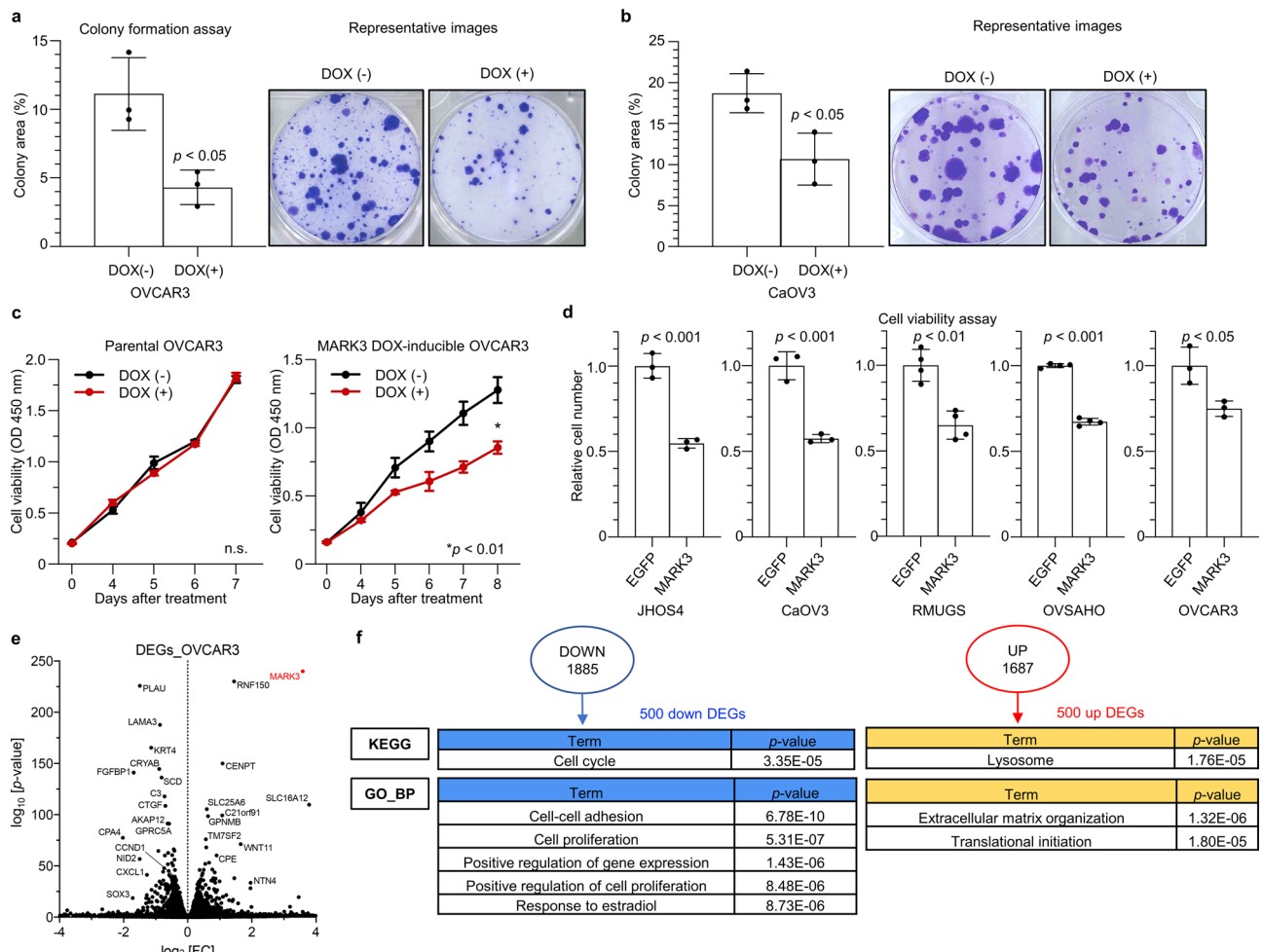

**Fig. 3 MARK3 overexpression inhibits cell proliferation in HGSOC cell lines. a**, **b** Colony formation assay in MARK3 doxycycline (DOX)-inducible OVCAR3 (**a**) and CaOV3 (**b**) cells. Error bars represent mean ± standard deviation (SD) of three biological replicates. Statistical analysis was performed using an unpaired Student's *t*-test. **c** Cell viability assay in parental OVCAR3 and MARK3 DOX-inducible OVCAR3. To exclude the potential cytotoxic effects of DOX, we treated both parental OVCAR3 and MARK3 DOX-inducible OVCAR3 cells with DOX and confirmed that only MARK3 DOX-inducible OVCAR3 showed a significant decrease in cell viability. Error bars represent mean ± SD of three biological replicates. Statistical analysis was performed using an unpaired Student's *t*-test. n.s.: not significant. **d** Cell viability assay for the transient overexpression of MARK3 in various types of HGSOC-like cell lines defined by genomic profiles, such as *TP53* mutation and hyper CNA. EGFP expression plasmid is used as a control. Relative cell number shows the relative ratio of the number of cells when EGFP and MARK3 are overexpressed, respectively (EGFP = 1.0). Error bars represent mean ± SD from three biological replicates. Statistical analysis was performed using an unpaired Student's *t*-test. **e** RNA-seq analysis. Volcano plot showing differentially expressed genes (DEGs) obtained by MARK3 overexpression in OVCAR3 cells of three biological replicates. **f** Results of the Kyoto Encyclopedia of Genes and Genomes (KEGG) pathway analysis and Gene Ontology (GO) analyses of the top 500 upregulated and downregulated DEGs. KEGG pathway analysis and GO analysis were performed on the top 500 downregulated and the top 500 upregulated DEGs separately. Downregulated DEGs are enriched in cell cycle pathway in KEGG pathways and cell-cell adhesion and cell proliferation in GO terms; upregulated DEGs are enriched in lysosome pathway in KEGG pathways and extracellular matrix organization and translational initiation in GO terms. A modified Fisher Exact *P*-value (EASE score) is shown.

As LKB1 is a well-characterized upstream activator of MARK3, it was further examined whether LKB1 is responsible for MARK3 activation upon exposure to metabolic stress. Indeed, LKB1 depletion abolished MARK3 activation under metabolic stress (Fig. 5c). Since anisomycin is known to activate JNK and p38 stress-activated kinases, the involvement of JNK and p38 in MARK3 activation under metabolic stress was evaluated. It was determined that p38 inhibitor, but not JNK inhibitor, attenuated MARK3 activation upon anisomycin treatment, suggesting that p38 also participated in the stress response to activate MARK3 (Supplementary Fig. 5b).

Analogous to synthetic lethality between DNA damage and CHEK1 inhibition[32], metabolic stress inducers might produce synthetic lethal effects upon MARK3 depletion. To this end, DOX-inducible MARK3-knockout TYK-nu cells, which harbored

HGSOC-like phenotype with an intact LKB1–MARK3 axis, were generated (Fig. 5d). Cell viability assays displayed that the cytotoxic effects of anisomycin and thapsigargin were significantly augmented upon MARK3 knockout (Fig. 5e, f). Overall, these results suggested that the LKB1–MARK3 axis was a cell cycle checkpoint activated by cytoplasmic metabolic stresses.

**MARK3 suppresses AP-1 and Hippo signaling target genes in HGSOCs**. Since known substrates of MARK3 cover a wide range of signaling pathways, such as RAS, cAMP-PKA, JNK, and Notch signaling as well as CDC25 signaling, the resultant cellular phenotype should be complicated by a mixture of these interactions. Thus, we consider that it is not appropriate to explain every cellular phenomenon only by CDC25 signaling. To obtain a comprehensive understanding of the biological function of

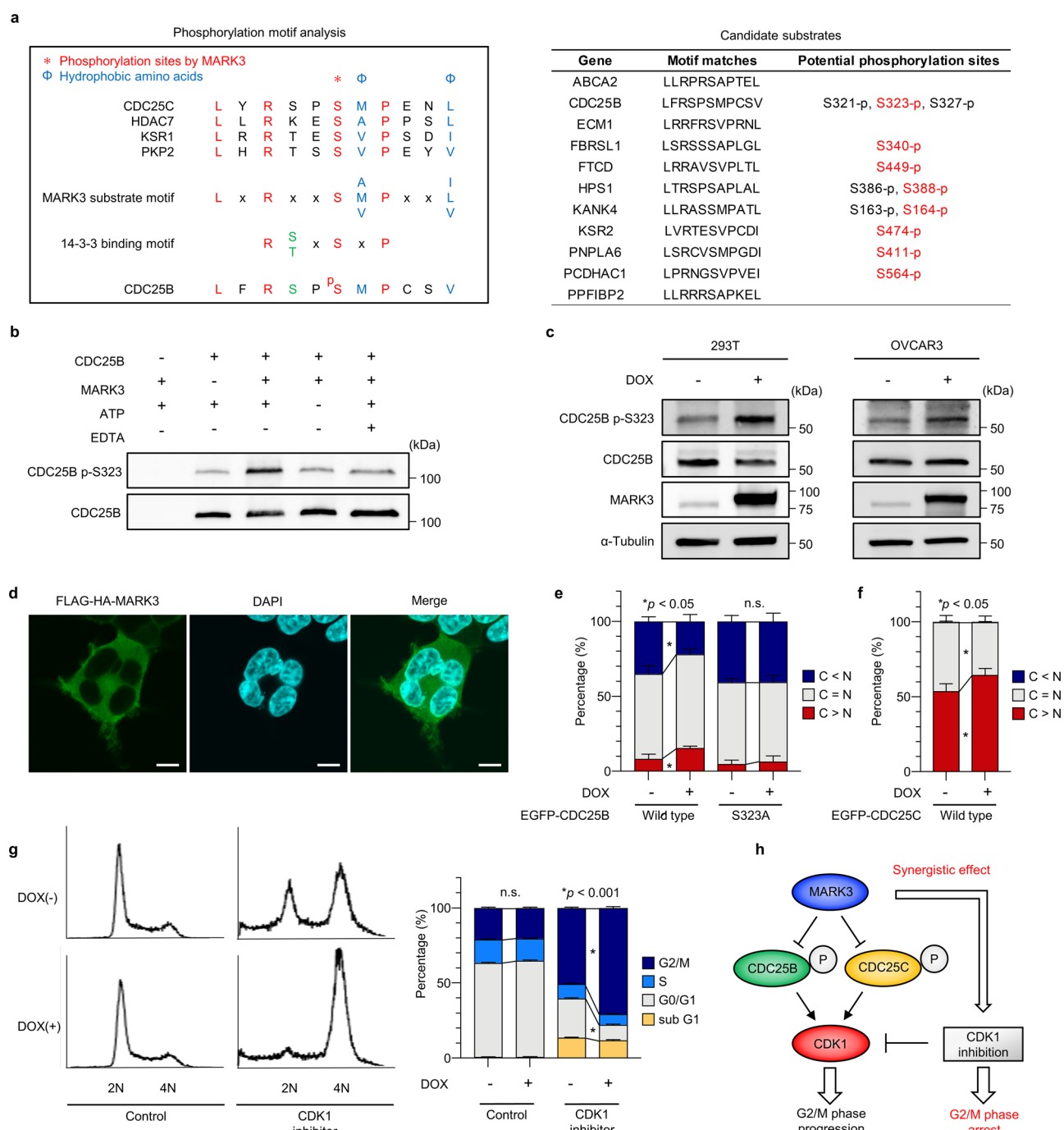

MARK3, ATAC-seq on MARK3-inducible OVCAR3 and 293T cells was performed. ATAC-seq is a sensitive method for assessing genome-wide chromatin accessibility and predicting the transcription factor (TF) binding profile[33]. By calculating the enrichment score of JASPAR core non-redundant motifs, it was predicted that MARK3 overexpression significantly retarded binding of the AP-1 complex, which is composed of *Jun* and *Fos* family members (Fig. 6a, b, and Supplementary Data 2, 3). Furthermore, decreased phosphorylation of c-Jun serine 63 upon MARK3 overexpression was observed, indicating that MARK3 acts as an antagonist of the AP-1 complex through the inhibition of c-Jun (Fig. 6c). Consistent with these results, the expression of AP-1 target genes, such as *PLAU, MMP1, CCND1,* and *CD44* decreased upon MARK3 overexpression (Fig. 6d)[21].

In addition, the integrative analysis of RNA-seq and ATAC-seq generated interesting findings. Our RNA-seq analysis revealed that MARK3 overexpression in OVCAR3 yielded downregulated genes enriched in the cell cycle, cell proliferation, and cell-cell adhesion pathways. Among them, *CTGF, CRYAB, FGFBP1,* and *PLAU,* which are well-characterized Hippo signaling target genes, were strongly downregulated. Hippo signaling is composed of tumor suppressors MST1/2 and LATS1/2, which regulate the activity of YAP/TAZ to suppress gene expression by the TF TEAD. YAP/TAZ are regarded as oncogenes that elevate TEAD transcriptional activity, promoting cell proliferation, angiogenesis, tumor invasion, and metastasis[34]. Intriguingly, recent studies reiterated that AP-1 and TEAD co-interact and synergistically promote the transcription of their target genes[35,36].

**Fig. 4 MARK3 phosphorylates CDC25B serine 323 and induces G2/M phase arrest. a** Overview of phosphorylation motif analysis is described in the left panel. We generated MARK3 substrate motif LxRxxS*[AVM]Pxx[ILV], where S* is potential phosphorylation site, x is optional, and brackets mean one of them, which were generated using amino acid sequences of known MARK3 substrates; CDC25C, HDAC7, KSR1, and PKP2. This potential MARK3 substrate motif coincides well with the known 14-3-3 binding motif R[ST]xSxP. Output data of candidate substrates are summarized in the right panel. Biologically conformed phosphorylation sites by mass spectrometry are listed. Potential phosphorylation sites by MARK3 are marked by red characters. **b** In vitro kinase assay. MARK3 increases the phosphorylation of CDC25B serine 323. This effect is reversed by adding EDTA or removing ATP. **c** Cell lysate assay. MARK3 overexpression in 293T and OVCAR3 cells increases the CDC25B p-S323 signal. **d** Immunocytochemistry shows that MARK3 is dominantly located in the cytoplasm and cell membrane in 293T cells. Anti-HA antibody was used to detect FLAG-HA-MARK3. Scale bars, 10 μm. **e** Ratio of subcellular localization of EGFP-tagged CDC25B proteins. 293T cells were transfected with EGFP-tagged CDC25B (wild type and S323A mutant type) expression plasmids with or without DOX treatment. MARK3 overexpression results in an increase in cytoplasmic-dominant localization and a decrease in nuclear-dominant localization of CDC25B. This phenomenon is not observed for the S323A mutant. Error bars represent mean ± standard deviation (SD) of three biological replicates. C < N: Nuclear-dominant. C = N: Equivalent. C > N: Cytoplasmic-dominant. Statistical analysis was performed using an unpaired Student's t-test. n.s.: not significant. **f** Ratio of subcellular localization of EGFP-tagged CDC25C proteins. 293T cells were transfected with EGFP-tagged CDC25C expression plasmids with or without DOX treatment. MARK3 overexpression results in an increase in cytoplasmic-dominant localization of CDC25C. Error bars represent mean ± SD of three biological replicates. Statistical analysis was performed using an unpaired Student's t-test. **g** Flow cytometry-based cell cycle analysis. Although MARK3 overexpression alone does not change the cell cycle distribution of untreated 293T cells, in response to CDK1 inhibitor treatment, it significantly increases the G2/M arrest as evidenced by the accumulation of the 4N population. Error bars represent mean ± SD of three biological replicates. Statistical analysis was performed using an unpaired Student's t-test. **h** Schematic of the effect of MARK3 inducing G2/M phase arrest with CDK1 inhibition.

To evaluate the effect of MARK3 on Hippo signaling, biologically validated Hippo signaling target genes were searched. Previously, Mohseni et al. reported a total of 177 Hippo signature genes, which are composed of DEGs in response to the inactivation of Hippo signaling by siRNA knockdown of NF2 and LATS2[37]. Zanconato et al. described a total of 379 YAP/TAZ direct targets confirmed by YAP/TAZ ChIP-seq and siRNA knockdown of YAP/TAZ[35]. These Hippo signature genes and YAP/TAZ direct targets include biologically validated Hippo signaling target genes such as *CTGF*, *CYR61*, and *ANKRD1*, confirming the reliability of gene selection.

Hippo signature genes and YAP/TAZ direct targets were adopted to DEGs obtained by MARK3 overexpression in OVCAR3. Strikingly, the volcano plot showed that these two gene sets were enriched in highly significantly downregulated DEGs (Fig. 6e, f). Indeed, immunoblotting showed that MARK3 overexpression decreased YAP transcriptional activity and diminished the protein expression of Hippo signaling target genes such as CTGF, CRYAB, and MYC (Fig. 6g). Supplementary Figure 6 shows the representative subcellular localization of YAP proteins when MARK3 is overexpressed.

Lastly, since highly downregulated DEGs such as *PLAU*, *CTGF* and *CRYAB* are known to promote angiogenesis in cancer, the in vivo tumor-suppressive effects of MARK3 were investigated via a mouse xenograft experiment using MARK3-inducible OVCAR3 cells. Subcutaneously transplanted mouse tumors (without interrupting the endogenous MARK3 expression) of MARK3 overexpressed OVCAR3 cells exhibited significantly smaller tumor growth than control tumors (Fig. 6h, i and Supplementary Fig. 7a). Furthermore, to evaluate the MARK3 effect on early phase angiogenesis, microvessel densities in tumors at 30 days from treatment were quantified by CD31 staining areas, and it was revealed that the DOX-positive group showed significantly less microvessel densities (Fig. 6j and Supplementary Fig. 7b, c). Taken together, our results indicated that MARK3 inhibited tumor cell proliferation and angiogenesis in ovarian cancer cells partly through the repression of AP-1 and YAP/TAZ/TEAD target genes.

## Discussion

In the present study, tumor-suppressive roles of the LKB1–MARK3 axis in HGSOCs were revealed. Our findings suggest that the molecules of the LKB1–MARK3 axis comprise the metabolic stress-activated checkpoint and are highly downregulated in HGSOCs, associated with poor clinical outcomes. *LKB1* (also known as *STK11*) is a well-known tumor suppressor gene; *LKB1* germline mutation causes Peutz–Jeghers syndrome, which is characterized by the development of intestinal hamartomas and a lifetime risk of multiple cancers, and *LKB1* somatic mutation has been observed in 17% of lung adenocarcinomas[38,39]. Although the significance of LKB1 in cancer biology is epidemiologically certain, the substrates of LKB1 that are required the most to suppress tumorigenesis remain unclear. The substrates of LKB1 are classified into five subfamilies: AMP-activated protein kinase (AMPK), brain-specific kinase (BRSK), SNF1-like kinase (NUAK), salt-inducible kinase (SIK), and MARK[12]. Among them, the LKB1–AMPK axis is widely studied and implicated in the control of cell metabolism, polarity, and growth[40]. In this study, anisomycin and thapsigargin activated the phosphorylation of MARK3 but not that of AMPK, suggesting that the activation mechanism of the LKB1–MARK3 axis is independent of the activation mechanism of the LKB1–AMPK axis (Fig. 5a). In addition, the LKB1–SIK axis has recently been described as a major tumor-suppressive pathway in non-small cell lung carcinomas[41]. Furthermore, as for the lack of enhancement of AMPKα phosphorylation after metformin administration (Fig. 5a), it has been pointed out in the previous reports that AMPK activation by metformin was not observed in the pharyngeal carcinoma cell line FaDu[42], and that in breast cancer cell lines, AMPK is activated in MCF-7 cells but not in MDA-MB-231 cells[43]. In the present study, metformin was administered at a relatively high concentration (1 mM) to avoid scenarios in which metformin did not work efficiently. These results suggest that rather than a lack of efficacy, the effect of metformin may vary based on the cellular environment of individual cells.

Importantly, the tumor-suppressive effects of the LKB1–MARK axis have not been elucidated. Rather, the MARK family of kinases (MARK1, MARK2, MARK3, and MARK4) are often reported as proto-oncogenic kinases with certain exceptions. For example, *MARK1* is amplified in various cancer types, such as breast and liver cancer, and *MARK2* is upregulated in lung cancer. However, these kinases have also been described as having tumor-suppressive effects[44–46]. *MARK4* is reportedly upregulated in liver cancer, gliomas, and glioblastomas[47,48]; however, the LKB1–MARK4 axis is reported to function as both an agonist and antagonist of Hippo signaling, suggesting cell-type dependent regulation of the LKB1–MARK4 axis[37,49].

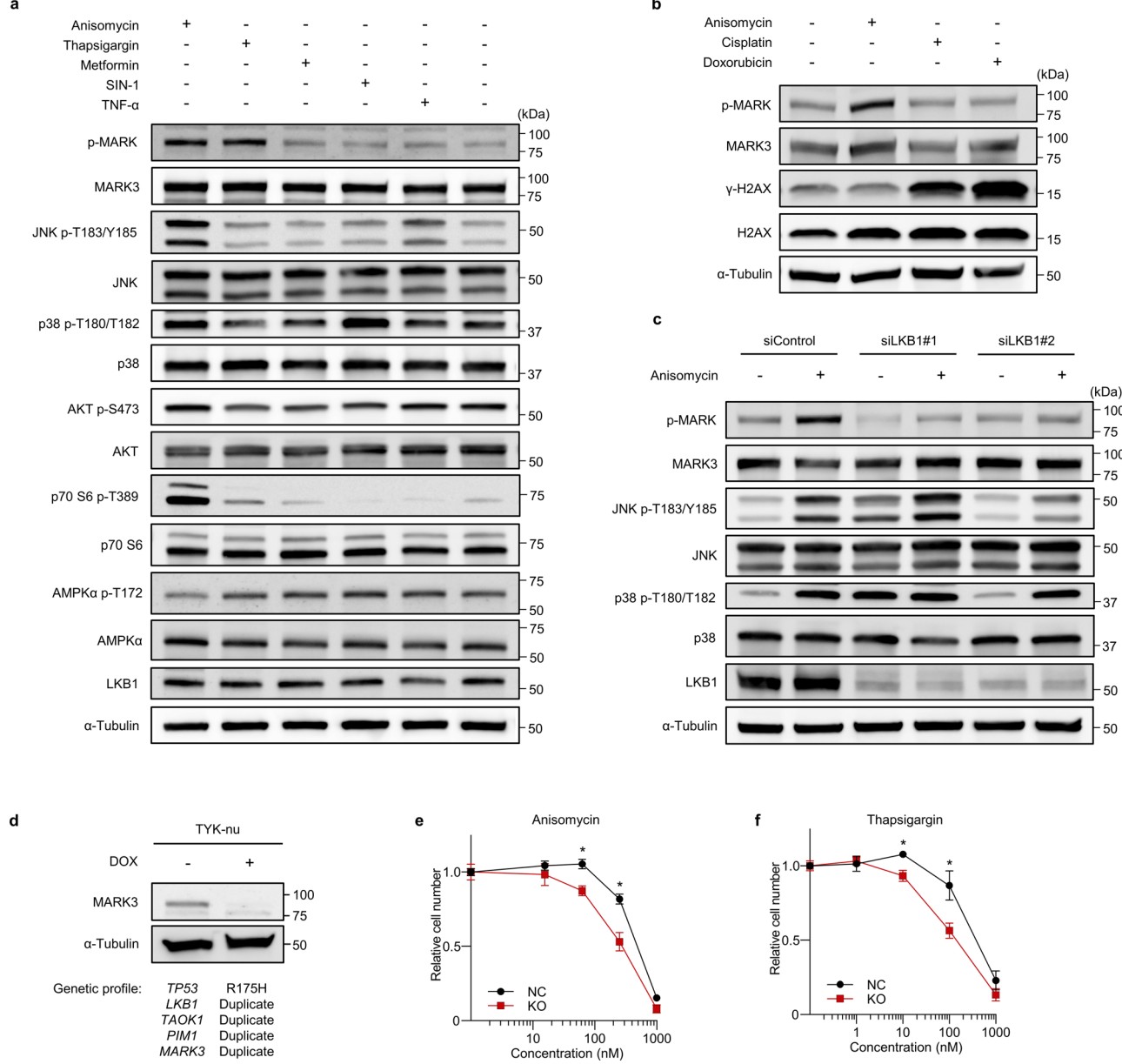

**Fig. 5 MARK3 is activated by metabolic stress inducers. a** Immunoblotting of chemical screening for metabolic stress inducers. Among tested compounds, anisomycin and thapsigargin increase the kinase-activated phosphorylation of doxycycline (DOX)-inducible MARK3. **b** Immunoblotting of chemical screening for DNA damage inducers with anisomycin as a positive control. DNA damage does not directly increase the phosphorylation of DOX-inducible MARK3. **c** Immunoblotting of inhibition experiments for the upstream regulators of MARK3. *LKB1* knockdown interferes with DOX-inducible MARK3 activation in response to anisomycin treatment. **d** CRISPR–Cas9 experiment for DOX-inducible *MARK3* knockout in TYK-nu cells. TYK-nu is a high-grade serous ovarian carcinoma (HGSOC)-like cell line, harboring the oncogenic *TP53* R175H mutation. The copy numbers of *LKB1*, *TAOK1*, *PIM1*, and *MARK3* are duplicates, suggesting that MARK3 can become active in TYK-nu. The genomic profile of TYK-nu is sourced from the Cancer Cell Line Encyclopedia (CCLE). **e**, **f** Cell viability assay for DOX-inducible *MARK3* knockout in TYK-nu. The cytotoxic effects of anisomycin (**e**) and thapsigargin (**f**) are amplified upon *MARK3* knockout. Relative cell number shows the relative ratio between the number of cells in the absence of the inhibitors (0 nM = 1.0). and the number of cells in the presence of various concentrations of anisomycin or thapsigargin. Error bars represent mean ± standard deviation (SD) of three biological replicates. Statistical analysis was performed using unpaired Student's *t*-test (Anisomycin 250 nM; Thapsigargin 100 nM; *$P < 0.01$).

Regarding *MARK3*, there is no consensus on whether it is an oncogene or a tumor suppressor gene. However, in contrast to the functions of the other MARK family kinases, the cancer-associated upregulation of *MARK3* is rarely reported. Besides, MARK3 is considered to antagonize various oncogenic pathways, such as the cell cycle[13], Ras signaling[23], cAMP-PKA signaling[24], JNK signaling[25], and Notch signaling pathways[26]. In particular, MARK3 can phosphorylate serine 216 of CDC25C, which pro-motes cytoplasmic translocation of CDC25C, resulting in G2/M

phase arrest[13]. This event is meaningful because DNA damage-activated checkpoint kinase CHEK1 also phosphorylates the same site of CDC25C in response to DNA damage[9]. The similarity between the functions of MARK3 and CHEK1 may indicate that MARK3 also functions as a cell cycle checkpoint kinase. In addition, our results showed that MARK3 phosphorylated serine 323 of CDC25B, which is a phosphorylation site of another checkpoint kinase, MAPKAPK2, to induce cytoplasmic translo-cation of CDC25B[10]. Since CDC25B/C are critical regulators of

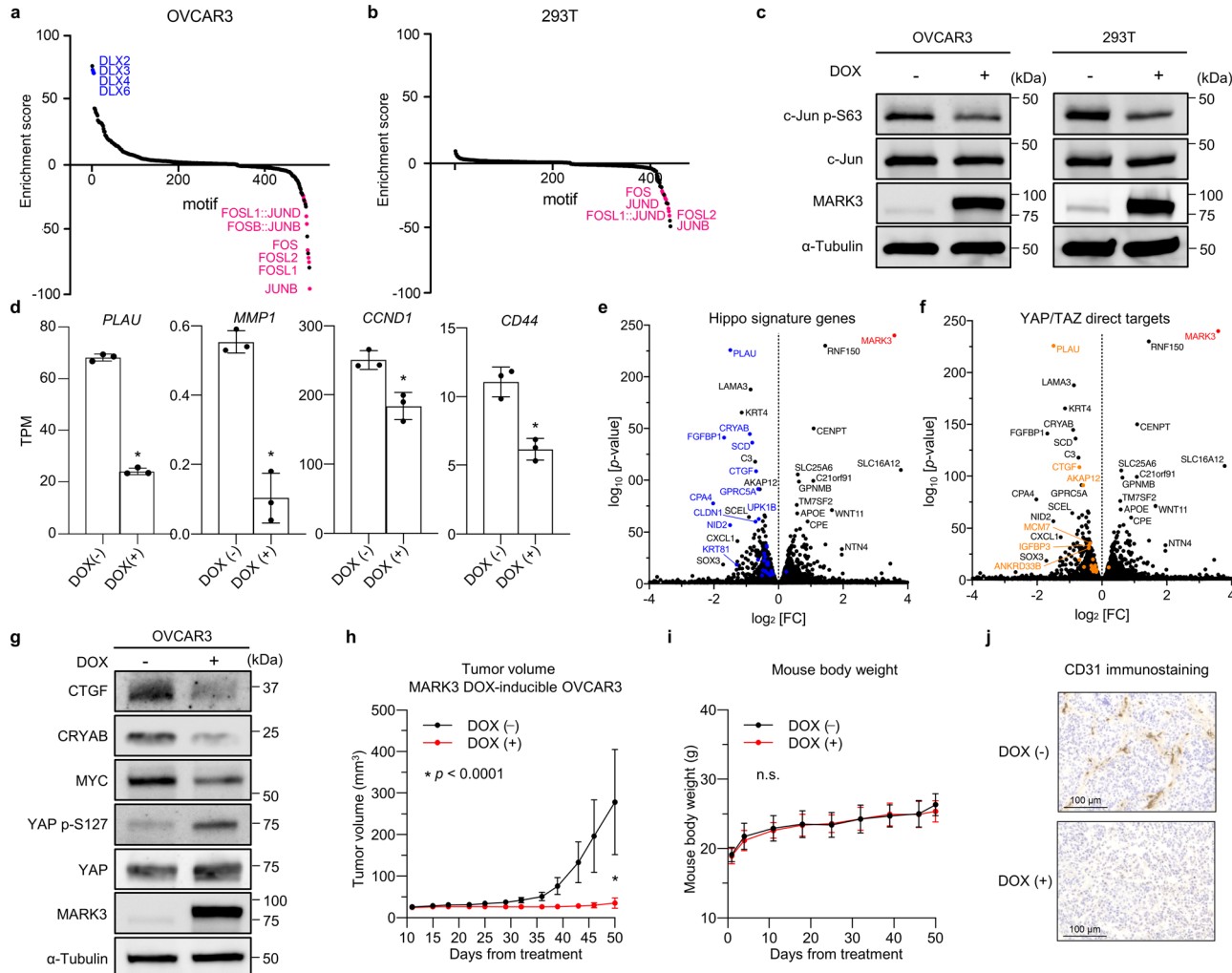

**Fig. 6 MARK3 regulate AP-1 and Hippo signaling target genes. a, b** ATAC-seq analysis. Ranking of the most differential motifs in response to MARK3 overexpression in OVCAR3 (**a**) and 293T (**b**) cells. In both cell lines, the activities of transcription factors, which make up the AP-1 complex, such as the Jun family and Fos family, are significantly decreased. Jaspar core non-redundant motifs are included in this motif enrichment analysis. **c** Immunoblotting shows that MARK3 overexpression in OVCAR3 and 293T cells decreases the phosphorylation of c-Jun, indicating that MARK3 antagonizes the activity of the AP-1 complex. **d** RNA-seq analysis. Representative AP-1 target genes are downregulated upon MARK3 overexpression. Error bars represent mean ± SD of three biological replicates. Statistical analysis was performed using unpaired Student's $t$-test (*$P < 0.01$). **e, f** RNA-seq analysis. Volcano plot demonstrates that MARK3 overexpression in OVCAR3 cells downregulates both Hippo signature genes (**e**) and YAP/TAZ target genes (**f**). **g** Immunoblotting shows that MARK3 overexpression in OVCAR3 cells inhibits nuclear translocation of YAP and decreases the protein expression of Hippo signaling target genes, such as CTGF and MYC. **h, i** Mouse xenograft experiment using MARK3 DOX-inducible OVCAR3 cells. The growth curves of the tumor volume (**h**) and the bodyweight (**i**) distribution of mouse xenografts. $n = 10$ in DOX-negative group and $n = 10$ in DOX-positive group. Error bars represent mean ± SD. Statistical analysis was performed using an unpaired Student's $t$-test. **j** Representative images of CD31 immunohistochemistry of mouse xenografts with or without DOX treatment.

the G2/M phase transition, our results collectively underscored the G2/M phase checkpoint activity of MARK3.

Furthermore, we observed an interesting contrast between CHEK1 and MARK3 in terms of subcellular localization and activation mechanism. CHEK1 is located in the nucleus and is activated by DNA damage; conversely, MARK3 is located in the cytoplasm and is activated by metabolic stress, such as ribosomal stalling and the unfolded protein response induced by anisomycin and thapsigargin, respectively. Indeed, our RNA-seq results indicated that MARK3 overexpression in OVCAR3 cells enhanced the expression of genes related to lysosomal and translational regulation, implying the involvement of MARK3 in the protein quality control response. There might be distinct cell cycle checkpoints to regulate CDC25C: the LKB1–MARK3 axis as a cytoplasmic metabolic stress-activated

checkpoint and the ATR–CHEK1 axis as a nuclear DNA damage-activated checkpoint.

Further comparisons between the LKB1–MARK3 axis and the ATR–CHEK1 axis can be made regarding their gene expression profiles. Since almost all HGSOCs are impeded by TP53 inactivation, the members of the ATR–CHEK1 axis tend to be upregulated to maintain genomic integrity (Supplementary Fig. 8). In contrast, the expression of *MARK3*, *LKB1*, as well as *TAOK1*, another upstream regulator of MARK3, was severely downregulated in HGSOCs. Considering these findings, it was postulated that HGSOCs inactivated the cytoplasmic G2/M phase checkpoint of the LKB1–MARK3 axis in order to neutralize the upregulated nuclear G2/M phase checkpoint of the ATR–CHEK1 axis (Supplementary Fig. 9).

The similarity between MARK3 and CHEK1 might offer therapeutic opportunities to patients with HGSOC. Since CHEK1 acts as a guardian of the DNA damage-activated G2/M phase checkpoint, inhibition of CHEK1 induces early entry into mitosis and causes mitotic catastrophe. Thus, the CHEK1 inhibitor is currently under clinical trial, showing an appreciable response to platinum and PARP-resistant HGSOCs[32]. Analogous to these results, it was hypothesized that the depletion of *MARK3* rendered cancer cells vulnerable to metabolic stress. Given that *MARK3*-knockout TYK-nu cells exhibited higher sensitivity to anisomycin and thapsigargin in our experiments, metabolic stress inducers might be beneficial for the treatment of LKB1–MARK3 axis-dysregulated HGSOCs.

Regarding the checkpoint activity of MARK3, Owusu et al. recently reported that the depletion of *MARK3* in HAP1 cells increased the genotoxic effects of DNA damaging agents[50]. Although the researchers only focused on the DNA damage response and did not evaluate the metabolic stress response, the study still offers valuable information supporting the possibility of MARK3 being a stress-activated checkpoint kinase. Importantly, *MARK3*-knockout HAP1 cells significantly decreased the number of phosphorylated proteins involved in the GO term of "regulation of translation". These data, in line with our results, coordinately suggest that MARK3 is a metabolic stress-activated checkpoint kinase involved in the protein quality control response.

Moreover, we attempted to observe the effect of MARK3 on cell cycle transition in OVCAR3 cells; however, we could not detect significant changes between DOX-negative and DOX-positive cells. This result may be attributed to the following two reasons. First, the suppressive effect on the phosphorylation of CDC25B in malignant ovarian cancer cells is strong, such that it is more difficult to detect G2/M phase cell cycle arrest in OVCAR3 than in 293T cells. Indeed, the western blotting results of OVCAR3 and 293T cells with or without DOX showed that the increased level of phospho-CDC25B caused by MARK3 over-expression in OVCAR3 was lower than that in 293T cells (Fig. 4c). Second, since MARK3 potentially affects many signaling pathways, such as RAS, cAMP-PKA, JNK, and Notch pathways, these interactions may complicate the resultant phenotype of cell cycle distribution. Our results suggest that the tumor-suppressive effects of MARK3 were derived not only from CDC25 signaling but also from AP-1 signaling, which reportedly caused G1/S phase arrest. Thus, it is assumed that there is a co-existence of G1/S phase arrest and G2/M phase arrest in MARK3-overexpressing OVCAR3 cells, making it difficult to detect each cell cycle arrest. However, we succeeded in conducting dose-dependent experiments using a combination therapy of CDK1 inhibitor and MARK3 overexpression in OVCAR3 cells. These results support the notion that MARK3 enhances the effect of CDK1 inhibition (Supplementary Fig. 4c).

Lastly, our research has certain limitations. Functional analysis was performed primarily using the means of rescue experiments in MARK3-inducible cell lines. However, given that HGSOCs arise from normal cells, such as HFTSEC or HOSE, the phenotypic changes occurring upon the dysfunction of the LKB1–MARK3 axis in normal cells should also be assessed. In this respect, Lennerz et al. reported that *MARK3*-knockout mice exhibit increased energy expenditure and reduced adiposity, where hepatic glycogen depletion and hypoketotic hypoglycemia are easily induced by overnight starvation[51]. From these results, we can presume that MARK3 potentially represses basal metabolism levels to protect cells from energy-consuming cellular processes. This is concordant with our conclusion that MARK3 acts as a metabolic stress-activated checkpoint kinase to inhibit the entry of cells into mitosis under metabolically stressful conditions. Additionally, it may be reasonable for cancer cells to diminish such MARK3 function to enhance metabolism and cell proliferation.

Given that HGSOCs suffer from the highest chromosomal instability, they inevitably produce both genomic and metabolic stress, such as the misfolded protein response caused by abnormal mRNA production[52]. It can be considered that the dysregulation of the LKB1–MARK3 axis enables HGSOCs to avoid a metabolic stress-activated checkpoint and potentiate their malignant phenotype. Synthetic lethal therapy using metabolic stress inducers, such as protein synthesis inhibitor or ER stress inducer might be beneficial for the treatment of LKB1–MARK3 axis-dysregulated cancer cells.

Overall, the present study suggested that the LKB1–MARK3 axis was a metabolic stress-activated G2/M phase checkpoint. The dysfunction of the LKB1–MARK3 axis in HGSOC can be a potential therapeutic target.

## Methods

**Ethics approval and consent to participate**. This study was approved by the Ethics Committee of National Cancer Center, Tokyo, Japan (approval ID: 2016-496) and the Human Genome, Gene Analysis Research Ethics Committee of University of Tokyo Hospital, Tokyo, Japan (approval ID: G0683-(18)), and by the Animal Experiment Committee of UNITECH (approval ID: AGR KGC-180216D-20) and the Genetic Rearrangement Experiment Safety Committee (approval ID: GR KGC-180216D-20). Clinical specimens were obtained with written informed consent from the donors. Mouse xenograft assays were conducted in accordance with the regulations of the Act on Welfare and Management of Animals, the Standards for the Care and Keeping of Laboratory Animals and Reduction of Pain, the Basic Guidelines of the MEXT, the Guidelines for Appropriate Animal Experimentation, and the Guidelines for Animal Disposal Methods.

**Cell lines and culture condition**. The 293T (expressing SV40 T-antigen), CaOV3, and ES2 cell lines were purchased from the American Type Culture Collection (ATCC, Manassas, VA, USA). JHOS-2, JHOS-4, and OVCAR3 cells were purchased from the RIKEN CELL BANK (Tsukuba, Japan). KURAMOCHI, OVSAHO, RMUGS, and TYK-nu cell lines were purchased from the Japanese Collection of Research Bioresources Cell Bank (JCRB, Osaka, Japan). 293T and CaOV3 cells were cultured in DMEM with 10% fetal bovine serum (FBS). JHOS-2 and JHOS-4 cells were cultured in DMEM/HamF12 medium with 10% FBS and 0.1 mM NEAA. ES2, KURAMOCHI, and OVSAHO cells were cultured in RPMI 1640 medium with 10% FBS, and OVCAR3 cells were maintained in RPMI 1640 medium with 20% FBS and 0.1% insulin. RMUGS cells were cultured in Ham's F12 medium with 10% FBS. TYK-nu cells were cultured in EMEM with 10% FBS. All cell lines were certified by STR profiling cell line authentication (Supplementary Data 4). We routinely confirmed that these cell lines were negative for mycoplasma contamination using an e-Myco mycoplasma PCR detection kit (25235; iNtRON Biotechnology, Kirkland, WA, USA). For the serum starvation study, cells were treated with 0.2% FBS for 24 h, and for the hypoxia study, cells were treated with 2% oxygen for 24 h.

**Clinical specimens**. The clinical specimens were frozen in liquid nitrogen immediately after sampling and stored at −80 °C. Tissue samples were embedded into the Optimal cutting temperature compound (OCT compound), followed by frozen sectioning and RNA extraction or immunohistochemistry.

**Bioinformatics analysis**. Two independent datasets (GSE18521 and GSE26712), containing microarray data of HOSE and HGSOC samples, were obtained from Gene Expression Omnibus (GEO). Differential expression analysis between HOSE and HGSOC was performed using the GEO2R pipeline, in which GEOquery and limma were utilized with default parameters. Gene expression levels were fitted to a log2 scale. RNA-seq, DNA promoter methylation, DNA copy number, gene mutation, and clinical data of HGSOC patients in The Cancer Genome Atlas (TCGA) cohort were sourced from the cBioPortal for Cancer Genomics. Survival curves were visualized by the Kaplan–Meier method and analyzed by the Log-rank test. Differential expression analysis between cancer and normal tissues across multiple cancer types was performed by gene expression profiling interactive analysis (GEPIA), in which RNA-seq data from TCGA and the Genotype-Tissue Expression (GTEx) projects were processed. To determine the cut-off between low/high expression, we applied three different ratios of 30%, 50%, and 70% to divide expression data into low/high groups. Among them, we found that the cut-off value, which divided 70% of samples into low and 30% of samples into high, yielded the highest statistical significance in the survival analyses for MARK3, BNC1, and MAF genes. Thus, we concluded that this cut-off was suitable for survival analysis in the present HGSOC cohort.

**Reverse-transcriptase quantitative real-time PCR (RT-qPCR)**. Total RNA from cell lines and clinical tissues was extracted using QIAzol Lysis Reagent and RNeasy Plus Mini Kit (73404; Qiagen, Valencia, CA, USA), and cDNA was synthesized using the PrimeScript RT Reagent Kit (RR037A; TaKaRa Bio, Kusatsu, Japan), according to the manufacturer's instructions. RT-qPCR reactions were performed using the TB Green Premix Ex Taq II (RR820A; TaKaRa Bio) and the CFX96 Touch system (Bio-Rad, Hercules, CA, USA). *MARK3* mRNA levels were normalized to *GAPDH* mRNA levels as an internal control using the ΔCq method. The detailed information of the primers used in this study is provided in Supplementary Data 5.

**Antibodies**. The following primary antibodies were used: Anti-α-Tubulin mouse monoclonal antibody (CP06 [EMD Millipore]; dilution used in WB: 1:1000); anti-β-Actin rabbit polyclonal antibody (#4967 [Cell Signaling Technology, Danvers, MA, USA]; dilution used in WB: 1:1000); anti-AKT rabbit monoclonal antibody (#4691 [Cell Signaling Technology]; dilution used in WB: 1:1000); anti-phospho-AKT (Ser473) rabbit monoclonal antibody (#4060 [Cell Signaling Technology]; dilution used in WB: 1:2000); anti-AMPKα rabbit polyclonal antibody (#2532 [Cell Signaling Technology]; dilution used in WB: 1:1000); anti-phospho-AMPKα (Thr172) rabbit monoclonal antibody (#2535 [Cell Signaling Technology]; dilution used in WB: 1:1000); anti-CD31 rat antibody (553370 [BD Biosciences]; dilution used in IHC: 1:500); anti-CDC25B rabbit monoclonal antibody (ab124819 [Abcam, Cambridge, UK]; dilution used in WB: 1:1000); anti-phospho-CDC25B (Ser323) rabbit polyclonal antibody (ab553103 [Abcam]; dilution used in WB: 1:300); anti-c-JUN rabbit monoclonal antibody (#9165 [Cell Signaling Technology]; dilution used in WB: 1:1000); anti-phospho-c-JUN (Ser63) rabbit polyclonal antibody (#9261 [Cell Signaling Technology]; dilution used in WB: 1:1000); anti-CRYAB rabbit monoclonal antibody (#45844 [Cell Signaling Technology]; dilution used in WB: 1:1000); anti-CTGF rabbit monoclonal antibody (#86641 [Cell Signaling Technology]; dilution used in WB: 1:1000); anti-GAPDH rabbit monoclonal antibody (#2118 [Cell Signaling Technology]; dilution used in WB: 1:1000); anti-HA mouse monoclonal antibody (901501 [BioLegend, San Diego, CA, USA]; dilution used in ICC: 1:1000 and in WB: 1:1000); anti-JNK rabbit polyclonal antibody (#9252 [Cell Signaling Technology]; dilution used in WB: 1:1000); anti-phospho-JNK (Thr183/Tyr185) rabbit monoclonal antibody (#4668 [Cell Signaling Technology]; dilution used in WB: 1:1000); anti-LKB1 rabbit monoclonal antibody (#3050 [Cell Signaling Technology]; dilution used in WB: 1:1000); anti-LKB1 rabbit monoclonal antibody (IHC Formulated) (#13031 [Cell Signaling Technology]; dilution used in IHC: 1:250); anti-MARK3 rabbit polyclonal antibody (#9311 [Cell Signaling Technology]; dilution used in WB: 1:1000); anti-MARK3 rabbit polyclonal antibody (ab133708 [Abcam]; dilution used in IHC: 1:100); anti-phospho-MARK family rabbit polyclonal antibody (#4836 [Cell Signaling Technology]; dilution used in WB: 1:1000); anti-p38 rabbit monoclonal antibody (#8690 [Cell Signaling Technology]; dilution used in WB: 1:1000); anti-phospho-p38 (Thr180/Tyr182) rabbit monoclonal antibody (#4511 [Cell Signaling Technology]; dilution used in WB: 1:1000); anti-p53 rabbit monoclonal antibody (#2527 [Cell Signaling Technology]; dilution used in IHC: 1:160); anti-p70 S6 rabbit monoclonal antibody (#2708 [Cell Signaling Technology]; dilution used in WB: 1:1000); anti-phospho-p70 S6 (Thr389) rabbit monoclonal antibody (#9234 [Cell Signaling Technology]; dilution used in WB: 1:1000); anti-PAX8 rabbit polyclonal antibody (10336-1-AP [Proteintech, Rosemont, IL, USA]; dilution used in IHC: 1:1000); anti-YAP rabbit monoclonal antibody (#14074 [Cell Signaling Technology]; dilution used in ICC: 1:100 and in WB: 1:1000); and anti-phospho-YAP (Ser127) family rabbit polyclonal antibody (#13008 [Cell Signaling Technology]; dilution used in WB: 1:1000). The following secondary antibodies were used: Anti-mouse IgG, horseradish per-oxidase (HRP)-linked species-specific whole antibody (from sheep) (NA931 [GE Healthcare]; dilution used in WB: 1:5000); anti-rabbit IgG, HRP-linked species-specific whole antibody (from donkey) (NA934 [GE Healthcare]; dilution used in WB: 1:5000); anti-mouse IgG (H + L) Alexa Fluor 488 (from donkey) (A21202 [ThermoFisher Scientific]; dilution used in ICC: 1:10000); EnVision+ System-HRP Labeled Polymer Anti-Rabbit (K4003 [Agilent Dako], an undiluted solution used in IHC); and biotinylated anti-rat IgG (H + L) (BA-9400 [VECTOR LABLATORIES], dilution used in IHC: 1:100).

**Immunoblotting**. Cells were directly lysed with the CelLyticM cell lysis reagent (C2978; Sigma-Aldrich, St. Louis, MO, USA), containing protease inhibitors (04693159001; Roche, Mannheim, Germany) and phosphatase inhibitors (39050; SERVA, Heidelberg, Germany). Whole-cell lysates were passed through a 25-gauge needle ten times before centrifugation. Total protein concentrations were measured using Pierce 660 nm Protein Assay Reagent (22660; ThermoFisher Scientific, Middletown, VA, USA). Whole-cell lysates mixed with Pierce Lane Marker Reducing Sample Buffer (39000; ThermoFisher Scientific) were boiled at 95 °C for 5 min, loaded into each lane of an SDS polyacrylamide gel (456-9034; Bio-Rad), followed by electrophoresis and transferred to a nitrocellulose membrane (10600012; GE Healthcare, Chicago, IL, USA). After blocking with 5% skim milk (190-12865; FUJIFILM Wako Pure Chemical Corporation, Tokyo, Japan), or the PhosphoBlocker Blocking Reagent (AKR-103, Cell Biolabs, San Diego, CA, USA) for the detection of phospho-proteins, membranes were incubated with primary antibodies at 4 °C overnight. Protein bands were marked by horseradish peroxidase (HRP)-conjugated antibodies and visualized by ECL Prime Western Blotting Detection Reagent (RPN2236; GE Healthcare) and ImageQuant LAS 4000 (GE Healthcare).

**Immunocytochemistry**. Cultured cells were fixed in 4% paraformaldehyde (163-20145; FUJIFILM Wako Pure Chemical Corporation) at 4 °C for 1 h, permeabilized in 0.1% Triton X-100 (160-24751; FUJIFILM Wako Pure Chemical Corporation) for 3 min, and blocked with 3% BSA (A9647; Sigma-Aldrich) at room temperature for 1 h. Fixed cells were incubated with primary antibodies at 4 °C overnight, followed by incubation with Alexa Fluor-conjugated secondary antibodies at room temperature for 1 h. For the detection of EGFP-tagged proteins, fixation and permeabilization steps were conducted as described above. After the addition of mounting medium with DAPI (H-1200; VECTOR LABORATORIES, Burlingame, CA, USA) on slide glasses, subcellular protein localization was evaluated using a BZ-9000 fluorescence microscope (Keyence, Osaka, Japan). The ratio of protein subcellular localization was calculated by counting at least 300 cells per group in three biological replicates.

**Immunohistochemistry**. For formalin-fixed paraffin-embedded tissue specimens, tissue sections were deparaffinized and antigen retrieval was performed under high pressure (110 °C, 5 min) in Target Retrieval Solution ×10 (S1699; Agilent Dako, Santa Clara, CA, USA). For fresh frozen tissue specimens, tissue sections were fixed in 4% paraformaldehyde (163-20145; FUJIFILM Wako Pure Chemical Corporation) at 4 °C for 10 min. Thereafter, tissues were incubated with 3% $H_2O_2$ (086-07445; FUJIFILM Wako Pure Chemical Corporation) diluted in methanol for 10 min to reduce endogenous peroxidase activity, followed by blocking with Blocking One Histo (06349-64; Nacalai Tesque, Kyoto, Japan) at room temperature for 10 min. Fixed tissue sections were incubated with primary antibodies at 4 °C overnight, followed by incubation with EnVision+ System- HRP Labeled Polymer Anti-Rabbit secondary antibodies (K4003; Agilent Dako) at room temperature for 30 min. For the detection of HRP reactions, EnVision DAB + Substrate Chromogen System (Agilent Dako, K3467) was used. For CD31 staining, a primary antibody of anti-CD31 rat antibody (553370; BD Biosciences, San Jose, CA, USA) and secondary antibody of biotinylated anti-rat IgG (H + L) (BA-9400; VECTOR LABLATORIES) were used, followed by peroxidase reaction using VECTASTAIN ABC kit (PK-4000; VECTOR LABLATORIES). Finally, tissue specimens were stained with Mayer's Hematoxylin Solution (30011; Muto Pure Chemicals, Tokyo, Japan) for 10 s to discriminate the nucleus from the cytoplasm. CD31-stained tissue areas were quantified using the BZ-X800 Analyzer software (Keyence).

**Plasmid DNA constructs**. The lentiviral packaging plasmids, pMD2.G and psPAX2 were obtained from Addgene (#12259 and #12260; Watertown, MA, USA). To generate inducible Cas9 nuclease-expression cell lines, Edit-R Inducible Lentiviral hEF1a-Blast-Cas9 Nuclease Plasmid DNA (CAS11229; GE Healthcare) was procured.

For the CRISPR–Cas9 knockout, sgRNA to target *MARK3* [5′–AGTCTGTAGTTTCCGATGTG–3′] was designed and cloned into pLKO.1-puro U6 sgRNA BfuAI large stuffer (#52628, Addgene).

To generate lentiviral vectors for conditional *MARK3* expression, a modified vector was constructed, using Edit-R Inducible Lentiviral hEF1a-Blast-Cas9 Nuclease Plasmid DNA as a backbone. To generate a unique restriction site, the NheI restriction site was mutated at the immediate downstream of the hEF1 promoter region using the Gibson assembly system (E2611; New England Biolabs, Ipswich, MA, USA). The N-terminal FLAG-HA-tagged *MARK3* cDNA was PCR amplified using human MARK3 expression plasmid (RC205758; OriGene, Rockville, MD), and cloned into the modified Edit-R Inducible Lentiviral Plasmid. The detailed information of the primers used in this study is provided in Supplementary Data 5.

A series of human *CDC25* family cDNAs were synthesized using gBlocks Gene Fragments (Integrated DNA Technologies, Coralville, IA, USA) and cloned into a pEGFP-N1 plasmid (6085-1, TaKaRa Bio) between XhoI/BamHI sites to produce C-terminal EGFP-tagged proteins. *MARK3* and *CDC25* family plasmids were verified by Sanger sequencing.

**Conditional protein expression and CRISPR–Cas9 knockout experiments**. A lentivirus transduction system was used to generate a conditional expression and CRISPR–Cas9 knockout cells. To produce lentiviruses, viral vector and packaging plasmids were co-transfected into the 293T using Lipofectamine 3000 (L3000-008; ThermoFisher Scientific), according to the manufacturer's instructions. After 48 h, the cell culture medium, containing lentiviruses for conditional FLAG-HA-MARK3, Cas9, or sgRNA expression, was collected and filtered through a 0.45-μm filter. Target cell lines were plated in 6-well plates and cultured with the lentivirus-containing medium for 3 days, which was carried out in the absence of polybrene. FLAG-HA-MARK3 or Cas9 conditional expression cells were selected with blasticidin S (10 μg/ml) (029-18701; FUJIFILM Wako Pure Chemical Corporation). For inducible CRISPR–Cas9 knockout experiments, conditional Cas9 expression cells further underwent lentivirus transduction of conditional sgRNA expression plasmid and selection in the presence of both blasticidin S (10 μg/ml) and puromycin (2 μg/ml) (ant-pr-1; InvivoGen, San Diego, CA, USA).

Conditional expression was induced by adding 1 μg/ml of doxycycline (DOX) (D9891; Sigma-Aldrich). For cell growth assays, a reduced amount of DOX (0.2 μg/ml) was used for conditional FLAG-HA-MARK3 expression to avoid the potential cytotoxic effect of DOX to the greatest possible extent.

**siRNA and plasmid DNA transfection**. siRNA and plasmid DNA transfections were performed using Lipofectamine RNAiMax transfection reagent (13778-150; ThermoFisher Scientific) and Lipofectamine 3000 reagent (L3000-008; Thermo-Fisher Scientific), respectively, following the manufacturer's instructions. Accu-Target Negative Control siRNA (SN-1013; Bioneer, Oakland, CA, USA) and MISSOIN *LKB1* siRNA (#1: SASI_Hs01_00092688 and #2: SASI_Hs01_00092689; Sigma-Aldrich) were used. For the cell viability assay, the indicated cell lines were transfected with the following expression vectors: MYC-DDK-tagged-MARK3 (RC205758; OriGene, Rockville, MD, USA) and pEGFP-N1 (6085-1; TaKaRa Bio).

**Colony formation assay**. Cells were plated in 6-well plates at the following concentrations; 2000 cells/well for CaOV3 and 3000 cells/well for OVCAR3. Culture media with or without DOX (0.2 μg/ml) were replaced every 3 days. CaOV3 and OVCAR3 cells were incubated for 25 and 30 days, respectively, until the colony staining procedure. Cells were fixed with 1% formaldehyde (252549; Sigma-Aldrich) and 1% methanol (137-01823; FUJIFILM Wako Pure Chemical Corporation), stained with 0.05% crystal violet (V5265; Sigma-Aldrich) for 20 min and then washed three times. Colony number and colony area were quantified using the ImageJ software.

**Cell viability assay**. Cells were plated in 96-well plates at the following concentrations: 3000 cells/well for CaOV3, OVCAR3, and TYK-nu; 5000 cells/well for JHOS-4, RMUGS, and OVSAHO. Culture media with the respective treatment agents were replaced every 3 days. At the indicated time points, 10 μl of Cell Counting Kit-8 (343-07623, Dojindo, Kumamoto, Japan) reagent was added to each well. After 2 h of reaction, cell viability was measured by detecting the absorbance at 450 nm using Multiskan FC (ThermoFisher Scientific).

**In vitro kinase assay**. Precisely 0.2 μg of recombinant human GST-tagged MARK3 (M45-10G; SignalChem, Richmond, BC, Canada) and a 1 μg of recombinant human GST-tagged CDC25B (SRP5006; EMD Millipore, Burlington, MA, USA) were incubated in the kinase buffer [25 mM Tris, 10 mM MgCl$_2$, 5 mM Glycerol-2-phosphate, 0.1 mM Na$_3$VO$_4$, 1 mM DTT, 0.1 mM ATP] at 30 °C for 60 min. For negative control experiments, kinase buffers with 0.05 M EDTA or without ATP were used to eliminate kinase activity. The reaction was terminated by adding SDS-PAGE sample buffer and boiling for 5 min. Samples were immediately processed for immunoblotting.

**Inhibitors and stress Inducers**. The following compounds were used under indicated conditions: Ro-3306 (CDK1 inhibitor, 9 μM, 20 h, S7747; Selleck Chemicals, Houston, TX); anisomycin (500 nM, 24 h, sc-3524; Santa Cruz Biotechnology, Dallas, TX, USA); thapsigargin (100 nM, 24 h, 586005; Sigma-Aldrich); metformin (1 mM, 24 h, 136-18662; FUJIFILM Wako Pure Chemical Corporation); SIN-1 (1 mM, 3 h, ab141525; Abcam); TNF-α (50 ng/ml, 24 h, 300-01 A; Pepro-Tech, Rocky Hill, NJ); cisplatin (2 μM, 24 h, S1166; Selleck Chemicals); doxorubicin (2 μM, 24 h, ab120629; Abcam); SP600125 (JNK inhibitor, 5 μM, 36 h, S1460; Selleck Chemicals); SB203580 (p38 inhibitor, 5 μM, 36 h, S1076; Selleck Chemicals).

**Mouse xenograft experiment**. Mouse xenograft experiments were performed in 8-week-old female BABL/cAJc1-nu/nu mice (CLEA Japan Inc., Tokyo, Japan). Based on weight measurements before injection, the mice were divided into two groups of five mice each so that the mean weight of each group such as DOX-positive (+) or DOX-negative (−) was approximately equal. MARK3 DOX-inducible OVCAR3 cells were precultured in a medium with 0.2 μg/mL DOX (D9891; Sigma-Aldrich) or the same volume of water, followed by subcutaneous injection of $1 \times 10^7$ cells in the left inguinal areas of mice. The DOX (+) group and DOX (−) control group mice were fed 5% sucrose liquid, containing 2 mg/ml DOX or the same volume of water, respectively. Tumors were resected at 50 days after subcutaneous injection. For CD31 immunostaining to evaluate early phase angiogenesis, additional mouse xenograft experiments were performed, and tumors were resected and paraffin-embedded at 30 days after subcutaneous injection. For tumor diameter measurement, the long and short diameters of the tumor masses were measured by the caliper, and the estimated tumor volume was calculated using the formula as follows:

$$tV = a \times b^2 \times 0.5 \text{ (tumor volume: tV, long diameter: } a, \text{ short diameter: } b)$$

The unit of tumor volume is mm$^3$; the unit for the long ($a$) and short ($b$) diameters of the tumor mass was mm, which was measured and calculated to one decimal place. For each of the five mice in the DOX (+) and DOX (−) groups, the mean tumor volume in the tumor mass was calculated, the unpaired Student's $t$-test by two-tailed distribution was used for the comparison of the two groups. Paraffin sections of the excised tissues were subjected to tissue immunohistochemistry using an anti-CD31 antibody, and the CD31 staining area was calculated using the BZ-X800 Analyzer software. Microscopically, the tumors often formed two or three independent masses separated by stroma, rather than a single mass. Therefore, in order to accurately calculate the CD31 staining area within the tumor tissue with the analysis software, each mass had to be cropped on the image data and the CD31 staining area had to be calculated individually. As a result, multiple masses were measured from a single slide, so we calculated the mean value for each slide and performed an unpaired Student's $t$-test by two-tailed distribution for the comparison of the two groups: DOX (+) group and DOX (−) group.

**RNA-seq and data analysis**. After the total RNA extraction and DNase I treatment, magnetic beads with Oligo (dT) were used to isolate mRNA. Mixed with the fragmentation buffer, the mRNA was fragmented into short fragments. Then cDNA was synthesized using the mRNA fragments as templates. Short fragments were purified and resolved with EB buffer for end reparation and single nucleotide A (adenine) addition. Subsequently, the short fragments were connected with adapters. The suitable fragments are selected for the PCR amplification as templates. During the QC steps, Agilent 2100 Bioanalyzer and ABI StepOnePlus Real-Time PCR System were employed in the quantification and qualification of the sample library. Finally, the library could be sequenced using Illumina HiSeq$^{TM}$4000 or another sequencer when necessary.

After sequencing, the raw reads were filtered. Data filtering includes removing adaptor sequences, contamination, and low-quality reads from raw reads. Next, the statistics of data production were obtained. The original image data are transferred into sequence data via base calling, which are defined as raw data or raw reads and saved as FASTQ files. FASTQ files are the original data provided for users, and they include the detailed read sequences and the read quality information.

RNA-seq reads were aligned to the human reference genome NCBI build hg38 using STAR. Transcripts per million transcripts (TPM) were calculated using RSEM, and DEGs were extracted using edgeR, in which FDR <0.05 were considered statistically significant. KEGG pathway analysis and GO analysis were processed on DAVID (v.6.8). Upstream regulator analysis was performed by ingenuity pathway analysis (IPA). To evaluate the influence of MARK3 overexpression on Hippo signaling target genes, previously described gene lists of Hippo signature genes and YAP/TAZ direct targets were utilized.

**ATAC-seq and data analysis**. ATAC-seq was performed by Active Motif. FASTQ files were processed for adapter sequence trimming, mapping to hg19 using bowtie2 using the option of -very sensitive -X 2000 and PCR duplicate removal. Mapping quality was assessed by DROMPA. Peaks were called by MACS2 using the options -f BAM -g hs -q 0.01 −nomodel −shift -75 −extsize 150 -B and further filtered with $P$-value < 10$^{-10}$. Peak raw counts were quantile normalized. Transcription factor (TF) motif enrichment analysis was performed as follows. A peak versus motif matrix was generated using HOMER, combining ATAC-seq peaks and JASPAR core non-redundant position frequency matrices on vertebrates. A peak versus motif matrix and a peak versus intensity matrix were integrated into the significance of the TF motif enrichment matrix by the Module Map algorithm of Genomica.

**Statistics and reproducibility**. Each experiment was repeated at least three times, and the experiments throughout the manuscript were successfully reproduced. Values are presented as mean ± standard deviation (SD). The unpaired Student's $t$-test by two-tailed distribution was used for the comparison of two groups. One-way analysis of variance (ANOVA) was used for the comparison of more than two groups. For the correlation analysis, Pearson's correlation coefficient was employed. A $P$-value of <0.05 was considered statistically significant unless otherwise specified.

**Reporting summary**. Further information on research design is available in the Nature Research Reporting Summary linked to this article.

## Data availability
Detailed values for individual data are provided in Supplementary Data 6. Untrimmed blots for immunoblotting are present in Supplementary Information (Supplementary Figs. 10–22). Sequence data generated in this study are available at the DNA Data Bank of Japan (DDBJ) (Accession number: DRA010685). The raw data and $t$-test results of the mouse xenograft experiments (Fig. 6h and Supplementary Fig. 7b) are shown in Supplementary Data 7.

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

## Acknowledgements

We are grateful to all members of Division of Molecular Modification and Cancer Biology in National Cancer Center Research Institute for technical support and helpful discussion. We also thank the members of Department of Obstetrics and Gynecology in Shimane University School of Medicine for sharing of immortalized cell lines. This work has been supported by Core Research for Evolutional Science and Technology (CREST) (Japan Science and Technology Corporation [JST]: Grant Number JPMJCR1689, to R.H.), Grant-in-Aid for Scientific Research on Innovative Areas (Japan Society for the Promotion of Science [JSPS]: Grant Number JP18H04908, to R.H.), and the subsidy for Advanced Integrated Intelligence Platform (Ministry of Education, Culture, Sports, Science and Technology [MEXT], to R.H.).

## Author contributions

Conceptualization, H.M., S.K., and R.H.; methodology, H.M., S.K., M.K., K.A., and R.H.; formal analysis, H.M., S.K., N.I., and R.H.; bioinformatics analysis of RNA-seq and ATAC-seq, H.M., S.K., and R.N.; in vitro and in vivo experimentation, H.M., S.K., and N.I.; sample preparation of RNA-seq and ATAC-seq, H.M., S.K., and N.I.; resources, K.S., H.Y., T.K., Y.O., and T.F.; original draft, H.M., S.K., and R.H.; writing—review and editing, H.M., S.K., M.K., N.I., K.A., R.N., K.Shozu, A.D., K.Sone, H.Y., T.K., K.O., Y.O., T.F., G.V.K., V.S., M.N., and R.H.; visualization, H.M., S.K., K.A., and R.H.; supervision, K.O., T.F., G.V.K., V.S., and R.H.; project administration, H.M., S.K., M.K., K.A., and R.H.; funding acquisition, R.H.

## Competing interests

The authors declare no competing interests.
