## [Transparent Peer Review File · Communications Biology]

Reviewers' comments:

Reviewer #1 (Remarks to the Author):

Title: The metabolic stress-activated checkpoint LKB1-MARK3 axis acts as a tumor suppressor in high-grade serous ovarian carcinoma.

Summary: Authors demonstrate a clear association between the downregulation of LKB1 and MARK3 in high-grade serous ovarian carcinoma (HGSOC) and the downstream implication of over-expression of MARK3 in inhibiting G2/M phase transition via CDC25B phosphorylation and inhibition of AP-1 and Hippo signaling in HGSOC cells. The integrative analysis is impressive. However, although the data support the association between LKB1 and MARK3 in gene-expression levels and copy number changes, the data pertaining to activation of this pathway in the context of metabolic stress is a bit weak. However, the overall clarity of the data and its impact could be improved if the comments are addressed.

Constructive Feedback:

1. Thapsigargin and anisomycin are not classic inducers of metabolic stress; instead, their mode-of-action is via protein synthesis perturbation. There are also no experiments in the manuscript that induce direct metabolic stress, such as nutrient starvation or hypoxia. Therefore, it is challenging to ascertain if the activation of LKB1-MARK3 is due to metabolic stress. Authors should test this hypothesis in various metabolic stress conditions and distinguish activation of LKB1-AMPK axis with LKB1-MARK3 axis in the context of metabolic stress.

2. Although the data on CDC25B phosphorylation by MARK3 is clear and robust, it is difficult to confer that CDK1 inhibition synergizes with MARK3 dependent phosphorylation of CDC25C and CDC25B or downregulation of AP1 signal. Additionally, the mechanism of how MARK3 over-expression reduced cell proliferation and colony formation is not clear. However, it can be tested in the present experimental models. There are clearly two mechanistic effects of MARK3 over-expression: 1, activation of inhibitory phosphorylation on CDC25C and CDC25B; 2, downregulation of AP1 and Hippo signaling. It is not clear from the data if these two processes are independent or connected. Authors should test the impact of CDK1 inhibitor on G2/M phase transition or cell proliferation in cells expressing a mutant form of CDC25B and overexpressing MARK3. It is essential to demonstrate whether the downstream impact of MARK3 is more dependent on CDC25B/C phosphorylation or AP1-regulation, or these two processes are connected. The overexpression of mutant CDC25B could also be used for rescue experiments in the colony formation assay. Authors can also use inhibitors of AP1 signaling in combination with thapsigargin or anisomycin in MARK3 downregulated HGSOC cells to show that if AP1 downregulation can synergize these cells to stress-induced by inhibitors of protein synthesis.

3. The inhibitors, anisomycin and thapsigargin, may also impact mTOR-dependent signaling. Authors should test if these inhibitor-dependent activations of LKB1-MARK3 axis have an impact on mTOR signaling or not.

4. The authors should show the growth curve of the tumors. Also, test whether CDC25B phosphorylation or downregulation of AP1 signal or both regulates tumor growth. mRNA and protein expression of the signaling components should be enough to direct to the mechanism.

Reviewer #2 (Remarks to the Author):

The authors in this study claim a new role for the LKB1-MARK3 axis as a metabolic stress activated tumor suppressive axis that phosphorylate cdc25 leading to G2/M arrest. Sequencing studies have been conducted to identify potential pathways and mechanisms downstream of MARK activation. The role of MARK's in phosphorylating Cdc25 is novel and well carried out as are the invitro characterization experiments on OVCAr3 cell behavior. The analysis of TCGA and published datasets and patient survival analysis are well controlled and well presented.

However, the rationale behind the ATAC seq (chromatin landscape) as a result of MARK alterations

and evaluation of angiogenesis in tumor studies that are underpowered is very unclear and does not provide clear evidence of any links between the cell cycle associated effects on the tumor cells as a result of MARK and the proposed vascular changes in the subcutaneous tumors. Specific questions and concerns are described below.

Figure 1: Changes to LKB1 in early STIC's and MARK3 in advanced tumors was seen. These observations are intriguing. Are these changes also manifested in a change in expression in a stage dependent manner- more specifically, in Fig 1 b,c what was the distribution of tumors that were early stage versus late. Can this data be provided?

Figure 3 and other functional assays: All studies were conducted using a Dox inducible MARK3. Additionally, Dox alone did not affect cell viability. However, given the availability of the FLAG tagged construct (Fig 4), the investigators would be well served to re-evaluate and validate some (not all) of the phenotypic consequences using a non-Dox overexpression. While the Dox inducible system is elegant, due to the stress induced response of MARK3 and the known effects of Dox on mitochondrial protein synthesis, non MARK dependent responses need to be ruled out.

Figure 4: It is not clearly described how the quantitation of cytoplasmic versus nuclear localization was conducted in d (not in method or legends). Examples of images should be provided. How were cells that express FLAG-HA selectively analyzed in 1e,f? – not described. Additionally, since no G2/M arrest was seen in 293 cells as a result of MARK expression, it is not clear what the consequence of the altered cdc25 localization is. Perhaps evaluating a more responsive cell line, may provide clearer evidence such as an ovarian cancer cell line. Additionally, authors have inaccurately stated the effect of CDK1 inhibition in MARK expressing cells, as synergy was not measured. Dose dependency experiments must be conducted to evaluate synergy.

ATAC seq: the goal of the ATAC seq is not clear- data along with the expression of the mutant MARK should be conducted to evaluate if the ATAC seq results are related to MARK's effects on cdc25. In the absence the conclusions are not tied to the findings in this manuscript- if they are – the authors need to elaborate better in the text.

RNAseq: MARK overexpression targets should be validated using a different overexpression system.

Tumor studies: It is not clear why angiogenesis was evaluated as no data to the effects on endothelial cells has been presented. What is the proposed mechanism of the vascular density changes?. Most importantly: Statistical power appears weak in these studies, only 5 mice per arm were used in an experiment with a broad output (tumor size ranging from 40-100 mm³). The experiment would require at least 10 mice per arm, and it needs to be repeated. There is also concern on the size of the tumors used for evaluating CD31 as tumors less than 50 mm³ are rather tiny and have been used for CD31 analysis. To evaluate angiogenesis accurately, tumors must be allowed to grow to a larger size, pictures of tumors can also be included. What would be most useful here, is the level of MARK in the tumors to know that the DOX has worked in vivo. In the absence of these the tumor studies are significantly underdeveloped.

Reviewer #3 (Remarks to the Author):

Communications Biology
Manuscript#: COMMSBIO-21-0105-T

The manuscript entitled "The metabolic stress-activated checkpoint LKB1-MARK3 axis acts as a tumor suppressor in high-grade serous ovarian carcinoma" focused on the impact of the LKB1-MARK3 axis in high grade serous carcinoma. Interestingly, the authors showed that a low MARK3 mRNA levels is

associated with a poor prognosis in HGSOC patients. The authors found that MARK3 mRNA expression is influenced by CNA. By using MARK3 inducible cellular models, the authors have shown that MARK3 overexpression decreases cell proliferation and angiogenesis through AP-1 and Hippo pathways. I have several major concerns which should be addressed before I can recommend this paper for publication.

Major concerns

1. In the figure 1, the authors identified 6 differentially expressed genes between normal ovarian tissue and HGSOC in 2 different data sets. Among them, they tested the overall survival and disease free survival of only 2 of them: BNC1 and MARK3.
 - a) How the cut off between low/high MARK3 or BNC1 has been established? This need to be indicated.
 - b) It should be interesting to test the 4 other genes, especially MAF expression with the patient clinical outcomes.
2. A paper from Zhang et al. in Cell (2016) provided proteomic data from 169 HGSOC tumors (TCGA). We recommend that the authors test the association between the level of the MARK3 protein and clinical outcomes. We thought it might be interesting to test whether the protein or the mRNA or both are indicative of the prognosis. Thus, it should be also important to test the correlation between mRNA and protein (in patient and in cell lines) before analyzing the CNA.
3. Concerning the hypothesis about the inactivation of MARK3, the authors mentioned the role of LKB1, TAOK1 and PIM1. The data presented in the current manuscript are not sufficient. We recommend to the authors :
 - a) The authors should correlate MARK3 mRNA levels with PIM1 mRNA levels in patients.
 - b) PIM1 silencing should be tested on MARK3 protein levels in vitro in ovarian cancer cells.
4. Proliferation rate should be investigated in high and low MARK3 HGSOC tumors, and in MARK3 inducible OVCAR3 xenograft mouse models .
5. How the authors explain the absence G2/M phase arrest upon MARK3 overexpression?
6. Phosphorylation of CDC25B (serine 323) should be tested upon CHEK1 inhibitor.
7. Concerning the molecular mechanism, the authors should test the activation of P-JNK and P-p38 upon anisomycin, thapsigargin, metformin, SIN-1 and TGF- α treatments. LKB1 protein level should also be investigated upon these treatments and upon p38 and JNK inhibitors. In addition, the activation of P-JNK and P-p38 upon LKB1 inhibition should be assessed.
8. YAP subcellular localization upon MARK3 overexpression is missing.

Minor concerns

1. The authors need to provide information about clinical data, such as the definition of "Sensitive" and "Resistant" in the figure 1f and also in the figure 1h.
2. GAPDH is not a loading control, especially between normal cells and cancer cells. The authors should provide another loading control for western blot analysis.
3. Concerning the epigenetic regulation assessed in the Supplementary Figure 3, the authors should precise which ovarian cancer cell line has been used.
4. It should be interested to test how MARK3 inhibition impacts the mRNA expression or the protein level of the other MARK proteins.

5. Phosphorylation of CDC25B should be investigated in OVCAR3 and CAOV3 inducible DOX cell lines. Images of the subcellular localization of CDC25B is missing and should be provided in figure 4.

Rebuttal letter

We are grateful for your consideration of our manuscript entitled “The metabolic stress-activated checkpoint LKB1-MARK3 axis acts as a tumor suppressor in high-grade serous ovarian carcinoma” (MS#: COMMSBIO-21-0105-T) by Machino *et al.* and appreciate your helpful comments.

Our responses to the reviewers are as follows:

[To reviewer #1]

Comment 1: Thapsigargin and anisomycin are not classic inducers of metabolic stress; instead, their mode-of-action is via protein synthesis perturbation. There are also no experiments in the manuscript that induce direct metabolic stress, such as nutrient starvation or hypoxia. Therefore, it is challenging to ascertain if the activation of LKB1-MARK3 is due to metabolic stress. Authors should test this hypothesis in various metabolic stress conditions and distinguish activation of LKB1-AMPK axis with LKB1-MARK3 axis in the context of metabolic stress.

Response 1: Thank you for your constructive feedback. Following your advice, we examined the effects of nutrient starvation (FBS reduction) and hypoxia in 293T cells overexpressing MARK3; no significant changes in the phosphorylation level of MARK3 were observed (Supplementary Fig. 5a in the revised manuscript). In addition, to verify whether anisomycin and thapsigargin affect the LKB1-AMPK axis, we checked the p-T172 levels of AMPK α and its loading control signals. Thus, neither anisomycin nor thapsigargin activated AMPK α , and we concluded that the activation mechanism of the LKB1-MARK3 axis is independent of that of the LKB1-AMPK α axis (Fig. 5a in the revised manuscript). The results of these new studies are detailed in the Results and Discussion sections of the revised manuscript (lines 276-285 [p12]; lines 369-372 [p16]).

Comment 2: Although the data on CDC25B phosphorylation by MARK3 is clear and robust, it is difficult to confer that CDK1 inhibition synergizes with MARK3 dependent phosphorylation of CDC25C and CDC25B or downregulation of AP1 signal. Additionally, the mechanism of how MARK3 over-expression reduced cell proliferation and colony formation is not clear. However, it can be tested in the present experimental models. There are clearly two mechanistic effects of MARK3 over-expression: 1, activation of inhibitory phosphorylation on CDC25C and CDC25B; 2, downregulation of AP1 and Hippo signaling. It is not clear from the data if these two processes are independent or connected. Authors should test the impact of CDK1 inhibitor on G2/M phase transition or cell proliferation in cells expressing a mutant form of CDC25B and overexpressing MARK3. It is essential to demonstrate whether the downstream impact of MARK3 is more dependent on CDC25B/C phosphorylation or AP1-regulation, or these two processes are connected. The overexpression of mutant CDC25B could also be used for rescue experiments in the colony formation assay. Authors can also use inhibitors of AP1 signaling in combination with thapsigargin or anisomycin in MARK3 downregulated HGSOC cells to show that if AP1 downregulation can synergize these cells to stress-induced by inhibitors of protein synthesis.

Response 2: Thank you for your constructive feedback. First, we performed a dose-escalation experiment using the CDK1 inhibitor Ro-3306 with and without MARK3 overexpression. The results revealed that MARK3 overexpression enhanced the cytotoxic effect of the CDK1 inhibitor (Supplementary Fig. 4c in the revised manuscript). Next, to verify whether this synergistic effect was derived from the inhibition of CDC25B by MARK3, we further performed experiments using mutant CDC25B (S323A). Unfortunately, the cytotoxic effect of lipofection itself, which is used during transfection of mutant CDC25B, is so strong that it significantly masks the tumor suppressive effect of the CDK1 inhibitor, and biologically meaningful results were not obtained. Instead,

the long-term effect of mutant CDC25B was confirmed by a colony formation assay, and the addition of mutant CDC25B negated the cyto-reductive effect of MARK3 overexpression, suggesting that CDC25B inhibition is essential for the tumor suppressive function of MARK3 (Supplementary Fig. 4d in the revised manuscript). Furthermore, when we attempted combination therapy with an AP-1 inhibitor (T-5224) and anisomycin or thapsigargin using OVCAR3 cells, no additive effect was observed under these conditions. Taken together, these results suggest that the tumor suppressive effect of MARK3 largely depends on its ability to phosphorylate CDC25B, at least in the *in vitro* setting. These results are described in the Results section of the revised manuscript (lines 259 [p11]-270 [p12]).

Comment 3: The inhibitors, anisomycin and thapsigargin, may also impact mTOR-dependent signaling. Authors should test if these inhibitor-dependent activations of LKB1-MARK3 axis have an impact on mTOR signaling or not.

Response 3: Thank you for your valuable advice. In response to your suggestion, we performed additional western blotting of AKT p-S473 (downstream of mTORC2) and p70 S6 p-T389 (downstream of mTORC1). The results showed that p70 S6 p-T389 was elevated by anisomycin (Fig. 5a in the revised manuscript). Because p70 S6 reportedly upregulates protein synthesis, its activation by anisomycin, which causes protein synthesis perturbation, seems to be biologically reasonable.

Comment 4: The authors should show the growth curve of the tumors. Also, test whether CDC25B phosphorylation or downregulation of AP1 signal or both regulates tumor growth. mRNA and protein expression of the signaling components should be enough to direct to the mechanism.

Response 4: In response to your comments, we repeated mouse xenograft experiments by extending the observation period and plotted a growth curve of the tumors (Fig. 6h in the revised manuscript). Owing to your constructive advice, our mouse xenograft

experiments yielded significantly improved results. However, due to the lack of antibodies for phosphorylation of CDC25B which is applicable for immunohistochemistry, we could not evaluate the phospho-protein levels during tumor formation.

Thank you so much for sharing your precious time with us. Our manuscript was significantly improved based on the constructive comments.

[To reviewer #2]

Comment 1: Figure 1: Changes to LKB1 in early STIC's and MARK3 in advanced tumors was seen. These observations are intriguing. Are these changes also manifested in a change in expression in a stage dependent manner- more specifically, in Fig 1 b,c what was the distribution of tumors that were early stage versus late. Can this data be provided?

Response 1: Thank you for your constructive feedback. We are also interested in the stage-dependent MARK3 expression. However, due to the aggressive phenotype of HGSOCs, we could not obtain adequate data for stage I HGSOC cases (for example, only one RNA-seq data in TCGA cohort). When we focused on stage II to stage IV cases, *MARK3* mRNA expression levels in stage II cases were slightly higher than those in stage III and stage IV cases (Rebuttal Fig. A).

Rebuttal Fig. A *MARK3* mRNA expression is slightly higher in stage II HGSOC than

stage III or IV HGSOc.

Comment 2: Figure 3 and other functional assays: All studies were conducted using a Dox inducible MARK3. Additionally, Dox alone did not affect cell viability. However, given the availability of the FLAG tagged construct (Fig 4), the investigators would be well served to re-evaluate and validate some (not all) of the phenotypic consequences using a non-Dox overexpression. While the Dox inducible system is elegant, due to the stress induced response of MARK3 and the known effects of Dox on mitochondrial protein synthesis, non MARK dependent responses need to be ruled out.

Response 2: Thank you for your feedback. Following your comments, we decided to test an additional non-DOX MARK3 overexpression system. To clarify our MARK3 overexpression system, we used two different MARK3 constructs: 1. FLAG-HA-tagged MARK3 was stably induced by DOX and 2. Myc-tagged MARK3 was transiently induced by lipofection. Because we already performed functional assays using FLAG-HA-tagged MARK3 in the DOX overexpression system, we used myc-tagged MARK3 to check the reproducibility of MARK3 activation under stress conditions. The results showed that myc-tagged MARK3 was also activated by anisomycin and thapsigargin (Supplementary Fig. 5a in the revised manuscript). Additionally, we added cell proliferation assay data of the transient MARK3 overexpression system and observed that myc-tagged MARK3 also showed cytoreductive effects in OVCAR3 and CaOV3 cells (Fig. 3d in the revised manuscript). Thus, it was deduced that the tumor suppressive effects of MARK3 are not limited to the conditions of the DOX overexpression system.

Comment 3: Figure 4: It is not clearly described how the quantitation of cytoplasmic versus nuclear localization was conducted in d (not in method or legends). Examples of images should be provided. How were cells that express FLAG-HA selectively analyzed in 1e,f? – not described.

Response 3: We apologize for the confusion, as there are a variety of epitope-tagged proteins. As shown in Fig. 4e and f, we transiently transfected EGFP-tagged CDC25B or EGFP-tagged CDC25C to visualize the autofluorescence of these proteins. Following your advice, we added example images of EGFP-tagged CDC25B, which included nuclear dominant, cytoplasmic dominant or equally distributed cells (Supplementary Fig. 4b in the revised manuscript).

Comment 4: Additionally, since no G2/M arrest was seen in 293 cells as a result of MARK expression, it is not clear what the consequence of the altered *cdc25* localization is. Perhaps evaluating a more responsive cell line, may provide clearer evidence such as an ovarian cancer cell line.

Additionally, authors have inaccurately stated the effect of CDK1 inhibition in MARK expressing cells, as synergy was not measured. Dose dependency experiments must be conducted to evaluate synergy.

Response 4: Thank you for your constructive feedback. Although we attempted to observe the effect of MARK3 on cell cycle transition in OVCAR3 cells, we could not detect significant changes in the simple comparison between DOX-negative and DOX-positive cells. There are two possible reasons for this result.

First, the suppressive pressure on the phosphorylation of CDC25B in malignant ovarian cancer cells is so strong that it is more difficult to detect G2/M phase cell cycle arrest in OVCAR3 than in 293T cells. Indeed, when we checked the western blotting results of OVCAR3 and 293T with or without DOX, the increased level of phospho-CDC25B by MARK3 overexpression in OVCAR3 was lower than that in 293T cells (Fig. 4c).

Second, since MARK3 potentially affects many signaling pathways, such as RAS, cAMP-PKA, JNK and Notch pathways, these interactions may complicate the resultant phenotype of cell cycle distribution. Our results suggest that the tumor suppressive effects

of MARK3 were derived not only from CDC25 signaling but also from AP-1 signaling, which reportedly caused G1/S phase arrest. Thus, it is assumed that there is a co-existence of G1/S phase arrest and G2/M phase arrest in MARK3 overexpressed OVCAR3 cells, making it difficult to detect each cell cycle arrest.

However, following your advice, we succeeded in conducting dose-dependent experiments using a combination therapy of CDK1 inhibitor and MARK3 overexpression in OVCAR3 cells. This result supports the notion that MARK3 augments the effect of CDK1 inhibition (Supplementary Fig. 4c in the revised manuscript).

Comment 5: ATAC seq: the goal of the ATAC seq is not clear- data along with the expression of the mutant MARK should be conducted to evaluate if the ATAC seq results are related to MARK's effects on cdc25. In the absence the conclusions are not tied to the findings in this manuscript- if they are – the authors need to elaborate better in the text.

Response 5: Thank you for this suggestion. Since known substrates of MARK3 cover a wide range of signaling pathways such as RAS, cAMP-PKA, JNK, and Notch signaling as well as CDC25 signaling, the resultant cellular phenotype should be complicated by a mixture of these interactions. Thus, we considered that it was not appropriate to explain every cellular phenomenon only by CDC25 signaling and were motivated to identify the most significant downstream influence of MARK3 using a comprehensive analytical tool. For this purpose, we utilized ATAC-seq, which is an emerging method for characterizing the transcription factor (TF) network.

ATAC-seq revealed that AP-1 signaling is a highly affected TF network. At the same time, we are aware that further investigation is needed to elucidate the direct molecular mechanism by which MARK3 interacts with AP-1 signaling. Nevertheless, the experimentally validated result of the deactivation of c-Jun by MARK3 overexpression ensures the quality of ATAC-seq data and may justify importance of AP-1 signaling as a potential therapeutic target for HGSOCS.

In response to your constructive comments, we included an additional explanation in the Results section of the revised manuscript (lines 303 [p13]-317 [p14]).

Comment 6: RNAseq: MARK overexpression targets should be validated using a different overexpression system.

Response 6: Thank you for your constructive feedback. In response to your suggestion, transient induction of myc-tagged MARK3 was performed in parental OVCAR3 cells. The results showed that mRNA expression of Hippo/AP-1 target genes such as *PLAU*, *CTGF* and *CRYAB* was not significantly changed. A possible explanation for this result could be that the cytotoxic effect of lipofection was so strong that it caused another stress response in OVCAR3 cells. Instead, to exclude the possibility of the side effects of DOX itself, we checked the mRNA expression of *PLAU*, *CTGF* and *CRYAB* in parental OVCAR3 cells upon DOX treatment and found no significant change under the conditions (Rebuttal Fig. B). We also added western blotting data showing downregulation of CTGF and CRYAB upon DOX-inducible MARK3 overexpression to validate the RNA-seq results (Fig. 6g in the revised manuscript).

Rebuttal Fig. B *PLAU*, *CTGF* and *CRYAB* mRNA expression levels with or without DOX treatment in parental OVCAR3 cells. No significant change is observed for these Hippo target genes in the condition of parental OVCAR3 cells. Error bars represent mean \pm standard deviation (SD) of three biological replicates. Statistical analysis was

performed using unpaired Student's t-test.

Comment 7: Tumor studies: It is not clear why angiogenesis was evaluated as no data to the effects on endothelial cells has been presented. What is the proposed mechanism of the vascular density changes?. Most importantly: Statistical power appears weak in these studies, only 5 mice per arm were used in an experiment with a broad output (tumor size ranging from 40-100 mm³). The experiment would require at least 10 mice per arm, and it needs to be repeated.

There is also concern on the size of the tumors used for evaluating CD31 as tumors less than 50 mm³ are rather tiny and have been used for CD31 analysis. To evaluate angiogenesis accurately, tumors must be allowed to grow to a larger size, pictures of tumors can also be included. What would be most useful here, is the level of MARK in the tumors to know that the DOX has worked in vivo. In the absence of these the tumor studies are significantly underdeveloped.

Response 7: Thank you for your constructive feedback. We focused on vascular densities in mouse tumors is because the significantly downregulated DEGs such as PLAU, CRYAB and CTGF reportedly promote tumor angiogenesis. To evaluate the early phase impact of MARK3 on angiogenesis, we performed CD31 staining in relatively low tumor volume samples at 30 days after treatment.

In response to your comments, we repeated mouse xenograft experiments extending the observation period to 50 days and plotted a growth curve for the tumors (Fig. 6h in the revised manuscript). Owing to your constructive advice, our mouse xenograft experiments yielded significantly improved results. However, due to the lack of antibodies applicable for immunohistochemistry of mouse-derived tissues, we could not evaluate protein expression levels in the present study.

Thank you so much for sharing your precious time with us. Our manuscript was significantly improved based on the constructive comments.

[To reviewer #3]

Comment 1a: In the figure 1, the authors identified 6 differentially expressed genes between normal ovarian tissue and HGSOC in 2 different data sets. Among them, they tested the overall survival and disease free survival of only 2 of them: BNC1 and MARK3. a) How the cut off between low/high MARK3 or BNC1 has been established? This need to be indicated.

Response 1a: Thank you for your constructive review. To determine the cut-off between low/high expression, we applied three different ratios of 30%, 50%, and 70% to divide expression data into low/high groups. Among them, we found that the cut-off value, which divided 70% of samples into low and 30% of samples into high, yielded the highest statistical significance in the survival analyses for MARK3, BNC1, and MAF genes. Thus, we concluded that this cut-off was suitable for survival analysis in the present HGSOC cohort. The details of the cutoff are described in the Methods section of the revised manuscript (lines 507-512 [p22]).

Comment 1b: b) It should be interesting to test the 4 other genes, especially MAF expression with the patient clinical outcomes.

Response 1b: Although not statistically significant (log-rank test $p = 0.06$), high MAF expression tended to lead to worse clinical outcomes. Thus, we have added this result to Supplementary Fig. 1f in the revised manuscript. The other three genes (*NKX3-1*, *PDE8B*, and *REEP1*) showed no reliable tendency (Rebuttal Fig. C).

Rebuttal Fig. C Kaplan-Meier survival curves. Kaplan-Meier survival curves classified by high or low *NKX3-1*, *PDE8B*, and *REEP1* mRNA expression in the TCGA HGSOC cohort.

Comment 2: A paper from Zhang et al. in Cell (2016) provided proteomic data from 169 HGSOC tumors (TCGA). We recommend that the authors test the association between the level of the MARK3 protein and clinical outcomes. We thought it might be interesting to test whether the protein or the mRNA or both are indicative of the prognosis. Thus, it should be also important to test the correlation between mRNA and protein (in patient and in cell lines) before analyzing the CNA.

Response 2: Thank you for sharing this information. Using proteomic data of HGSOC, we confirmed the high correlation between mRNA and protein expression levels of MARK3 (Supplementary Fig.1d in the revised manuscript). As for the clinical outcome, low protein expression of MARK3 was slightly associated with worse progression-free survival in HGSOC patients, although it needs adjustment of the cut-off value (Rebuttal Fig. D).

Rebuttal Fig. D Kaplan–Meier survival curves. Kaplan–Meier survival curves classified

by high ($n = 81$) or low ($n = 73$) MARK3 protein expression in the TCGA HGSOC cohort.

Comment 3: 3. Concerning the hypothesis about the inactivation of MARK3, the authors mentioned the role of LKB1, TAOK1 and PIM1. The data presented in the current manuscript are not sufficient. We recommend to the authors :

a) The authors should correlate MARK3 mRNA levels with PIM1 mRNA levels in patients.

b) PIM1 silencing should be tested on MARK3 protein levels in vitro in ovarian cancer cells.

Response 3: Thank you for your constructive comment. Using the TCGA HGSOC cohort, we found no significant correlation between *MARK3* mRNA expression and *PIM1* mRNA expression as well as *LKB1* and *TAOK1* mRNA expression. Functionally, the PIM1-MARK3 axis is a kinase cascade; therefore, PIM1 may not directly affect MARK3 protein expression levels. Since PIM1-MARK3 interaction is another important topic in stress responses, we will investigate the function of this pathway in a future research project.

Comment 4: Proliferation rate should be investigated in high and low MARK3 HGSOC tumors, and in MARK3 inducible OVCAR3 xenograft mouse models.

Response 4: Thank you for your constructive feedback. In response to your comments, we repeated mouse xenograft experiments by extending the observation period and created a growth curve for the tumors (Fig. 6h in the revised manuscript). Owing to your constructive advice, our mouse xenograft experiments yielded significantly improved results.

Comment 5: How the authors explain the absence G2/M phase arrest upon MARK3 overexpression?

Response 5: Thank you for your constructive comments. Although we focused on the

interaction of the MARK3 and CDC25 family in the present study, MARK3 potentially interacts with numerous signaling pathways, such as RAS, cAMP-PKA, JNK, and Notch pathways. In addition, we also observed that AP-1 signaling, which reportedly progressed to the G1/S phase transition, was antagonized by MARK3. Thus, we assume that these diverse functions of MARK3 may affect not only the G2/M phase but also other cell cycle checkpoints, masking the potential G2/M phase arrest upon MARK3 overexpression.

Comment 6: Phosphorylation of CDC25B (serine 323) should be tested upon CHEK1 inhibitor.

Response 6: Thank you for your constructive comments. We observed that the CHEK1 inhibitor AZD7762 diminished the phosphorylation level of CDC25B serine 323 in 293T cells (Rebuttal Fig. E). We believe that this result is interesting, but the specificity of the CHEK1 inhibitor (AZD7762) needs to be carefully evaluated. We also need to carefully discuss the relationship between CHEK1 and phosphorylation of CDC25B serine 323; thus, for the present, we have decided not to include the results in the manuscript.

Rebuttal Fig. E Effect of AZD7762, a CHEK1 inhibitor, on the phosphorylation of CDC25B serine 323.

Comment 7: Concerning the molecular mechanism, the authors should test the activation of P-JNK and P-p38 upon anisomycin, thapsigargin, metformin, SIN-1 and TGF- α treatments. LKB1 protein level should also be investigated upon these treatments and

upon p38 and JNK inhibitors. In addition, the activation of P-JNK and P-p38 upon LKB1 inhibition should be assessed.

Response 7: Thank you for your constructive comments. Following your advice, we added western blotting data for these signaling pathways (Fig. 5a,c, and Supplementary Fig. 5b in the revised manuscript).

Comment 8: YAP subcellular localization upon MARK3 overexpression is missing.

Response 8: Thank you for your constructive comments. YAP subcellular localization highly depends on cell confluency, causing a strong bias in the cell counting processes. Thus, we did not calculate the proportion of subcellular localizations. However, we prepared a representative image of YAP subcellular localization in MARK3 DOX-inducible OVCAR3 cells (Supplementary Fig. 6 in the revised manuscript).

Comment 9 (minor): The authors need to provide information about clinical data, such as the definition of “Sensitive” and “Resistant” in the figure 1f and also in the figure 1h.

Response 9: Thank you for your constructive comments. The clinical data and the definitions used in Fig. 1f and 1h were originally posted by the TCGA project. We understand that the definition of platinum-sensitive ovarian cancer follows a universal consensus: “ovarian cancer that comes back 6 or more months after platinum-based treatment is considered platinum sensitive.”(<https://www.cancer.gov/publications/dictionaries/cancer-terms/def/platinum-sensitive-cancer>).

Comment 10 (minor): GAPDH is not a loading control, especially between normal cells and cancer cells. The authors should provide another loading control for western blot analysis.

Response 10: Thank you for your constructive comments. We agree with your concern

regarding loading control between normal cells and cancer cells. Initially, we checked the expression of α -tubulin, but noticed that its expression dynamically differs between each cell type. Thus, we used GAPDH (Fig. 1i) because its expression was relatively stable across normal and cancer cells. However, following your suggestion, we added β -Actin as another loading control and found that its expression was quite stable in all cell types (Fig. 1i).

Comment 11 (minor): It should be interested to test how MARK3 inhibition impacts the mRNA expression or the protein level of the other MARK proteins.

Response 11: We apologize for the failure to specify the name of the cell line and appreciate your kind notification. CaOV3 cells were used in this experiment because deletion of the MARK3 locus was not reported for CaOV3 cells. We have added a description of the cell lines in the figure and the figure legend of Supplementary Fig. 3a in the revised manuscript.

Comment 12 (minor): It should be interested to test how MARK3 inhibition impacts the mRNA expression or the protein level of the other MARK proteins.

Response 12: Thank you for your constructive comments. When the expression of MARK family mRNA was surveyed using TCGA data, no significant correlation was observed for each family protein.

Comment 13 (minor): Phosphorylation of CDC25B should be investigated in OVCAR3 and CAOV3 inducible DOX cell lines. Images of the subcellular localization of CDC25B is missing and should be provided in figure 4.

Response 13: Thank you for your constructive comments. We added the western blotting data demonstrating the increased phosphorylation level of CDC25B serine 323 upon MARK3 overexpression in OVCAR3 cells (Fig. 4c in the revised manuscript). We could

not detect a visible change in CaOV3 samples, possibly because the amount of overexpressed MARK3 in CaOV3 was lower than that in OVCAR3. In addition, following your advice, we added example images of EGFP-tagged CDC25B, which included nuclear dominant, cytoplasmic dominant, or equally distributed cells (Supplementary Fig. 4b in the revised manuscript).

Thank you so much for sharing your precious time with us. Our manuscript was significantly improved based on the constructive comments.

REVIEWERS' COMMENTS:

Reviewer #1 (Remarks to the Author):

I commend authors to address my comments in the previous version of the manuscripts. The additional work has added clarity to the manuscript and improved the overall impact of the work. I have no further comments.

Reviewer #2 (Remarks to the Author):

The authors have addressed all the issues raised by this reviewer which has significantly improved the understanding of this paper. One minor suggestion is to incorporate some of the explanations in the rebuttal letter directly into the discussion to facilitate understanding to a reader as well.

Reviewer #3 (Remarks to the Author):

The manuscript entitled "The metabolic stress-activated checkpoint LKB1-MARK3 axis acts as a tumor suppressor in high-grade serous ovarian carcinoma" focused on the impact of the LKB1-MARK3 axis in high grade serous carcinoma. This revised manuscript includes additional data that aligns much better with the conclusions. In general, the authors have adequately addressed the previous concerns. As it is now, the manuscript is more compelling and the results are better supported by experimental evidence. Overall, this is a far better study, now suitable for publication in Communications Biology in my opinion.

There are, however, some points that need to be clarified/addressed.

Comment 1: From our previous comment 4, we still have questions about the proliferation capacity. In the xenograft experiments presented in Figure 6h, no difference in tumor growth is observed until 50 days (the images in Supplementary Figure 7a are very convincing), whereas in Supplementary Figure 7b, a significant difference is observed at 30 days, not reported/observed in Figure 6h. How do the authors explain these different results?

Comment 2: It's very interesting to see the different tests performed to induce metabolic stress, such as hypoxia and nutrient starvation as well as the molecular read out tested by the authors. But in the revised version of the manuscript, no information is given for the time the cells are placed in hypoxia and at which O₂%, time of FBS starvation, etc. The lack of activation of AMPK α is surprising after metformin treatment. The authors should discuss these points and clarify them in the manuscript.

Comment 3: Experiment done with CHEK1 inhibitor should be at least discussed (and may be added) in the revised manuscript, as it is indicated in the Supplementary Fig. 9 Schematic of the signal transduction systems around the LKB1-MARK3 axis.

Rebuttal letter

We are grateful for your consideration of our manuscript entitled “The metabolic stress-activated checkpoint LKB1-MARK3 axis acts as a tumor suppressor in high-grade serous ovarian carcinoma” (MS#: COMMSBIO-21-0105A) by Machino *et al.* and appreciate your helpful comments.

Our responses to the reviewers are as follows:

[To reviewer #1]

Comment: I commend authors to address my comments in the previous version of the manuscripts. The additional work has added clarity to the manuscript and improved the overall impact of the work.

I have no further comments.

Response 1: Thank you so much for sharing your precious time with us. Our manuscript was significantly improved based on the constructive comments.

[To reviewer #2]

Comment 1: The authors have addressed all the issues raised by this reviewer which has significantly improved the understanding of this paper. One minor suggestion is to incorporate some of the explanations in the rebuttal letter directly into the discussion to facilitate understanding to a reader as well.

Response 1: Thank you for your constructive feedback. In response to your suggestion, we have included part of the explanations in the rebuttal letter in the Discussion section of the revised manuscript (lines 443-460).

Thank you so much for sharing your precious time with us. Our manuscript was significantly improved based on the constructive comments.

[To reviewer #3]

Comment 1: From our previous comment 4, we still have questions about the proliferation capacity. In the xenograft experiments presented in Figure 6h, no difference in tumor growth is observed until 50 days (the images in Supplementary Figure 7a are very convincing), whereas in Supplementary Figure 7b, a significant difference is observed at 30 days, not reported/observed in Figure 6h. How do the authors explain these different results?

Response 1: Thank you for the constructive review. In Figure 6h, the asterisk is shown only at day 50 for visibility, but in reality, significant differences ($p < 0.05$) were observed at all observation time points after day 18. In Figure 6h, $p = 1.03E-6$ at day 32, and in Supplementary Figure 7b, $p = 0.03$ at day 30. The second mouse experiment was significantly better than the first experiment on day 30. Since $n = 5$ for the first time and $n = 10$ for the second time, there was a tendency for significant differences to appear in the second time; thus it is believed that the first and second times generally reflect similar trends. The data and t-test results of Figure 6h and Supplementary Figure 7b are shown in Supplementary Data 7 in the revised manuscript.

Comment 2: It's very interesting to see the different tests performed to induce metabolic stress, such as hypoxia and nutrient starvation as well as the molecular read out tested by the authors. But in the revised version of the manuscript, no information is given for the time the cells are placed in hypoxia and at which O₂%, time of FBS starvation, etc. The lack of activation of AMPK α is surprising after metformin treatment. The authors should discuss these points and clarify them in the manuscript.

Response 2: Thank you for your constructive feedback. For the serum starvation study, cells were treated with 0.2% FBS for 24 h, and for the hypoxia study, cells were treated with 2% oxygen for 24 h. We have described these conditions in the Methods section of the revised

manuscript (lines 512-514). Additionally, since metformin has been reported to activate AMPK by increasing AMP levels, the reviewer is right in pointing out that the lack of enhanced phosphorylation of AMPK α following metformin administration, is surprising. On the other hand, it has been pointed out that the results differ depending on the cell environment, such as the fact that metformin did not activate AMPK in the pharyngeal carcinoma cell line FaDu (DOI: 10.1074/jbc.M116.769141) and that AMPK was activated in the breast cancer cell line MCF-7 but not in MDA-MB-231 (DOI: 10.1158/0008-5472.CAN-07-2310). In this experimental system, metformin was administered at a relatively high concentration (1 mM) to avoid scenarios in which metformin did not work efficiently. These results suggest that rather than a lack of efficacy, the effect of metformin may vary based on the cellular environment of individual cells. We have described these findings in the Discussion section of the revised manuscript (lines 374-381).

Comment 3: Experiment done with CHEK1 inhibitor should be at least discussed (and may be added) in the revised manuscript, as it is indicated in the Supplementary Fig. 9 Schematic of the signal transduction systems around the LKB1-MARK3 axis.

Response 3: Thank you for your feedback. We have previously explained this point; however, studies in the past have shown that CHEK1 phosphorylates serine 216 of CDC25C and not serine 323 of CDC25B. In Supplementary Figure 9, the signaling pathway of CHEK1 \rightarrow CDC25C is described, while the signaling pathway of CHEK1 \rightarrow CDC25B is not described. Therefore, the additional experiment presented in the previous revision (attenuation of serine 323 of CDC25B by CHEK1 inhibitor) and Supplementary Fig. 9 are not strictly related to each other, and we consider that it would be difficult to add an explanation in the text.

Thank you so much for sharing your precious time with us. Our manuscript was significantly improved based on the constructive comments.